# Role of Adipose-Derived Mesenchymal Stem Cells in Bone Regeneration

**DOI:** 10.3390/ijms25126805

**Published:** 2024-06-20

**Authors:** Chau Sang Lau, So Yeon Park, Lalith Prabha Ethiraj, Priti Singh, Grace Raj, Jolene Quek, Somasundaram Prasadh, Yen Choo, Bee Tin Goh

**Affiliations:** 1National Dental Centre Singapore, National Dental Research Institute Singapore, Singapore 168938, Singapore; gmslauc@nus.edu.sg (C.S.L.); park.so.yeon@ndcs.com.sg (S.Y.P.); ethiraj.lalith.prabha@ndcs.com.sg (L.P.E.); grace.raj@ndcs.com.sg (G.R.); 2Oral Health Academic Clinical Programme, Duke-NUS Medical School, Singapore 169857, Singapore; 3Lee Kong Chian School of Medicine, Nanyang Technological University, Singapore 308232, Singapore; quek0226@e.ntu.edu.sg (J.Q.); yen.choo@ntu.edu.sg (Y.C.); 4Center for Clean Energy Engineering, University of Connecticut, Storrs, CT 06269, USA; somasundaram.prasadh@uconn.edu

**Keywords:** adipose-derived stem cells, bone regeneration, bone tissue engineering, regenerative medicine

## Abstract

Bone regeneration involves multiple factors such as tissue interactions, an inflammatory response, and vessel formation. In the event of diseases, old age, lifestyle, or trauma, bone regeneration can be impaired which could result in a prolonged healing duration or requiring an external intervention for repair. Currently, bone grafts hold the golden standard for bone regeneration. However, several limitations hinder its clinical applications, e.g., donor site morbidity, an insufficient tissue volume, and uncertain post-operative outcomes. Bone tissue engineering, involving stem cells seeded onto scaffolds, has thus been a promising treatment alternative for bone regeneration. Adipose-derived mesenchymal stem cells (AD-MSCs) are known to hold therapeutic value for the treatment of various clinical conditions and have displayed feasibility and significant effectiveness due to their ease of isolation, non-invasive, abundance in quantity, and osteogenic capacity. Notably, in vitro studies showed AD-MSCs holding a high proliferation capacity, multi-differentiation potential through the release of a variety of factors, and extracellular vesicles, allowing them to repair damaged tissues. In vivo and clinical studies showed AD-MSCs favoring better vascularization and the integration of the scaffolds, while the presence of scaffolds has enhanced the osteogenesis potential of AD-MSCs, thus yielding optimal bone formation outcomes. Effective bone regeneration requires the interplay of both AD-MSCs and scaffolds (material, pore size) to improve the osteogenic and vasculogenic capacity. This review presents the advances and applications of AD-MSCs for bone regeneration and bone tissue engineering, focusing on the in vitro, in vivo, and clinical studies involving AD-MSCs for bone tissue engineering.

## 1. Introduction

### 1.1. Bone and Bone Injuries

Bone is a living tissue responsible for several important functions in the body, such as providing structural support, facilitating body movements, and providing protection for vital organs. Several physiological processes, such as homeostasis, endocrine functions, and hematopoiesis [1,2,3], also occur in bone, making it a vital organ for regulation. Bone fractures are among the most frequent organ injuries, and can be the result of a direct trauma or other pathological conditions such as cysts or the post-debridement of infections or tumors [4]. Usually, bone tissues exhibit a high self-repair and regeneration capacity through the recruitment of osteoprogenitor cells from their surroundings [5]. However, in the event of critically sized bone injuries, the damage exceeds the natural healing capacity leading to delayed healing, scar formation, and persistent bone defects. This often requires external intervention in order for healing to occur [6]. Often, limited or impaired vascularization and a significant loss of progenitor cells underlie these conditions, possibly worsened by the patient’s existing condition(s), lifestyle, or genetic factors [7].

### 1.2. Current Treatments for Bone Injuries

Bone is the second-most transplanted tissue in the world (after blood) and the treatment of bone defects costs the US USD 5 billion annually [8]. The high prevalence of bone injuries constitutes the need for therapeutic interventions to promote vascular and tissue formation with the aim of restoring the function of the defect site [9,10]. Currently, the gold standard for bone regeneration and defect reconstruction is bone grafting, where autologous bone tissue is transplanted to bridge the gap in the defect zone. However, drawbacks such as the donor site morbidity, deficiency in the amount of transplanted tissue, immunological issues, and graft resorption restrict the efficacy of these treatments and have motivated the need for alternative solutions [11,12]. An alternative approach involves the use of allogenic bone grafts or synthetic bone substitutes, which mitigates the complications associated with harvesting and quantitatively limited graft materials [13,14]. However, the use of allogenic grafts carries the risk of immunogenic rejection due to limited tissue matching and the possibility of disease transmission [15,16]. Synthetic bone substitutes, on the other hand, lack sufficient osteoinductive and osteogenic properties, unable to facilitate optimal osteoconduction while simultaneously controlling the resorption time and biomechanical support required for large defects [17]. Nevertheless, the use of grafts poses risks and its limitations instigate further investigation for better techniques to facilitate bone regeneration.

### 1.3. Cell-Based Treatments

Bone tissue engineering is a promising approach to overcome the limitations of existing grafts. It involves combining living cells and biomaterials, together with biochemical and physical factors to create tissue-like structures with the aim of providing sufficient nutrients, oxygen, and structural support in the defect cavity for bone formation [18,19,20,21]. In the ideal scenario, cells seeded into the biocompatible scaffold and within the fracture zone would undergo further osteogenic differentiation, the secretion of osteogenic factors, and the recruitment of osteoblast progenitor cells [18]. The interwoven structures of the scaffold would then support the growth of cells and tissues, eventually forming bone.

Cells used for bone tissue engineering should possess the following characteristics: (1) convenient harvesting and little trauma to the body; (2) good biological activity and an osteogenic ability; (3) a strong in vitro amplification ability; and (4) low toxicity, good biocompatibility, no tumorigenicity, etc [22]. Stem cells, which are unspecialized cells with the capability to self-renew and differentiate into various cell types [23], possess most of the characteristics mentioned above and are, therefore, increasingly applied in bone tissue engineering. There are four main types of stem cells [24,25,26]:Adult stem cells—also known as somatic stem cells, these undifferentiated cells can be found within specific differentiated tissues.Embryonic stem cells (ESCs)—ESCs originate from the inner cell population of the blastocyst embryo and have unlimited self-renewal and the ability to differentiate into other cell lines from the three germ layers.Extra-embryonic stem cells—these cells are derived from extra-embryonal sources, the primary source being isolated from tissue discarded after birth. These strains have a good multilineage differentiation ability.Induced pluripotent stem cells (iPSCs)—these pluripotent stem cells are derived from adult stem cells by reprogramming with inducing genes and factors.

### 1.4. Mesenchymal Stem Cells

Among the four main types of stem cells, adult stem cells are the most commonly used in tissue engineering due to their abundance in the body and fewer ethical constraints. In particular, mesenchymal stem cells (MSCs), which are adult stem cells with the ability to differentiate into specialized cells from the mesoderm, become important in cell therapy and regenerative medicine due to their self-renewal ability, multipotent differentiation ability, easy accessibility, and exceptional genomic stability [27,28].

MSCs can be isolated from a variety of tissues, such as bone marrow, adipose tissue, umbilical cords, amniotic fluid, skin, dental pulp, synovium, and ovarian follicular fluid [29,30]. Bone marrow-derived mesenchymal stem cells (BM-MSCs), obtained from the bone marrow cavity during surgery or during bone marrow aspiration, are the most frequently used MSCs in bone tissue engineering, as BM-MSCs were the first MSCs identified and their high osteogenic differentiation potentials have been extensively characterized [31,32]. Despite their effectiveness in osteogenic differentiation, BM-MSCs have the following disadvantages: a low cell yield from bone marrow aspirate, a decreasing differentiation ability with increasing age, a painful and invasive harvesting procedure, and the need for lengthy cell expansion in a clean facility to achieve sufficient cell numbers [33,34].

The limitations of BM-MSCs prompted researchers to explore alternative and dependable sources of MSCs. In the early 2000s, adipose tissue was discovered independently by Zuk et al. [31,35] and Halvorsen et al. [36] as a source of multipotent stem cells capable of multiple mesodermal lineage differentiation including adipocytes, osteoblasts, and chondrocytes [37]. Halvorsen et al. [36] demonstrated that subcutaneous adipose tissue can be easily accessed through liposuction, potentially yielding up to one billion cells per 300 g of adipose tissue. The discovery of these adipose-derived mesenchymal stem cells (AD-MSCs) revolutionized the field of regenerative medicine, as AD-MSCs have a number of advantages over BM-MSCs, such as a high abundance of adipose tissue within the body, easier accessibility of adipose tissues, a less invasive extraction procedure, a higher cell yield, and a higher proliferation rate [38,39]. AD-MSCs, when seeded into biomaterials and implanted into defect sites, have shown promising bone regeneration [18] and have demonstrated the capacity to undergo osteogenesis rapidly with minimal stimulation by exogenous cytokines [40]. These advantages render AD-MSCs to be an attractive alternative source for bone regeneration.

In this review, we exclusively investigated the role of AD-MSCs in bone regeneration, reporting the characteristics of AD-MSCs and its unique properties that makes it a valuable source of somatic stem cells with osteogenic potential. This article thus aims to validate the capability of AD-MSC as an excellent candidate for bone tissue engineering by reviewing the techniques and advantages of using AD-MSCs reported from studies published in the past decade. We will also suggest possible strategies to further enhance the potential of AD-MSCs as a state-of-the-art bone regenerative tool and recommend future research directions.

## 2. From Fat to Bone: Achieving Bone Regeneration with AD-MSCs

### 2.1. Adipose Tissue as Source of Stem Cells

Adipose tissue, a multifaceted organ with structural, endocrinal, energy-storing, and immunomodulatory functions, is present in all mammalian species as well as some non-mammalian species [41,42]. Adipose tissue contains a heterogeneous population of cells comprising mature adipocytes (>90%) and a stromal vascular fraction (SVF) containing stromal/stem cells (15–30%), endothelial cells (10–20%), pericytes (3–5%), and hematopoietic cells such as granulocytes, monocytes, and lymphocytes (25–45%) [43,44]. The amount of stem cells in adipose tissue varies according to age, anatomical location, and histotype. White adipose tissue is more abundant and serves to protect other organs and store excess energy, while brown adipose tissue, so named because of its high mitochondria content, serves to generate body heat [41]. AD-MSCs capable of multi-lineage differentiation can be found in white adipose tissue and they are more abundant in subcutaneous fat deposits than in visceral fat deposits [45].

Adipose tissue is derived from the mesoderm, along with other connective tissues such as the dermis, cartilage, bone, and the circulatory system [41]. The mesodermal origin of AD-MSCs enable them to differentiate into adipogenic, angiogenic, cardiomyogenic, chondrogenic, myogenic, osteogenic, periodontogenic, and tenogenic lineages induced by lineage-specific induction factors [45]. Despite their mesodermal origin, AD-MSCs have also been shown to differentiate into cells of ectodermal and endodermal origins [46].

### 2.2. Isolation and Osteogenic Differentiation of AD-MSCs

Although AD-MSCs can be found in various adipose depots in the body, subcutaneous adipose tissue, especially from the abdomen, is the most popular location to obtain AD-MSCs due to its easy and repeatable access, as well as the greater yield of AD-MSCs derived from subcutaneous depots [47]. The buccal fat pad (BFP) has gained attention in recent times, proving to be an attractive option for the reconstruction of small bony defects in the craniofacial region. It is a vascularized adipose tissue located between the oral cavity and the outer cheek, allowing ease of harvest, is less invasive, and has minimal donor site morbidity [48].

Adipose tissue reserved for AD-MSC isolation can be harvested as a solid tissue via surgical excision or harvested as a liquid aspirate via liposuction, where adipose deposits are detached from the adipose depot’s fibrous network and aspirated out of the body using a vacuum pump or syringe [49,50]. Among the various protocols to isolate and process AD-MSCs, the method described by Zuk et al. [35] is still the most established and most widely used. In this method, the harvested adipose tissue is washed extensively with phosphate-buffered saline (and finely minced if the tissue is in solid form) before being subjected to collagenase digestion at 37 °C for 30–60 min. After neutralizing the collagenase with culture medium containing 10% fetal bovine serum, the mixture is filtered and centrifuged to yield a pellet containing the stromal vascular fraction (SVF) which is a heterogenous population of cells including AD-MSCs, endothelial cells, and immune cells. AD-MSCs can then be isolated from the SVF by incubating the SVF in plastic flasks/plates overnight and retaining/expanding the plastic-adherent cells.

The most reported method to differentiate AD-MSCs into the osteogenic lineage is to culture them in an osteogenic medium formulated using a standard base media (e.g., Dulbecco’s modified Eagle’s medium (DMEM) or Minimum Essential Medium Alpha (α-MEM)) supplemented by sera (e.g., fetal bovine serum), inductive factors (e.g., dexamethasone), ascorbates (e.g., ascorbic acid or ascorbate-2-phosphate), and phosphates (e.g., β-glycerophosphate) [51]. The osteogenic differentiation of AD-MSCs can be further enhanced by soluble factors such as bone morphogenetic proteins (BMPs), vitamin D3, platelet-rich plasma, human platelet lysate, selenium [52], or inorganic ionic products such as calcium [53,54], hydroxyapatite [55], strontium [56], or magnesium [57]. The osteogenic differentiation of AD-MSCs is also influenced by the surface properties of the scaffold or extracellular matrix, such as stiffness [58], porosity [59], fiber alignment [60], and chemical composition [61,62], as well as external stimuli such as the shear [63,64], electromagnetic field [65,66], and acoustic waves [67,68].

### 2.3. Dedifferentiated Fat Cells

Stem cells can also be obtained from adipose tissue by isolating mature adipocytes by a “ceiling culture” method and inducing the mature adipocytes into lipid-free multipotent cells [69]. The “ceiling culture” method is used because unlike most cell types, mature adipocytes float and adhere to the upper wall of the culture flask during culturing. When induced, the mature adipocytes gradually shed lipid droplets and turn into a fibroblast-like state [70]. Stem cells obtained this way are known as dedifferentiated fat (DFAT) cells. It is important to distinguish between DFAT cells and AD-MSCs when referring to adipose stem cells, as AD-MSCs are adult stem cells while DFAT cells are a form of induced pluripotent stem cells. Studies to compare DFAT cells and AD-MSCs for bone regeneration showed that both DFAT cells and AD-MSCs demonstrated mesenchymal stem cell characteristics and both have similar osteogenic potentials [71,72], and DFAT cells have been proven to be effective for bone regeneration in a rat calvarial model [73,74]. Despite showing positive scientific outcomes and offering great promises, DFAT cells are still not as widely used as AD-MSCs due to a number of obstacles and challenges such as cell culture purity, phenotypic properties, and dedifferentiation mechanisms [75]. Therefore, this review will be mainly focusing on the use and applications of AD-MSCs.

### 2.4. The Mechanism of Bone Formation and Bone Repair

The process of bone formation, also known as osteogenesis or ossification, begins during the first few weeks of embryonic development and continues throughout childhood and adulthood [76]. In adults, bone development continues to occur in the form of bone remodeling and bone fracture repair. Bone formation starts with cell migration and a mesenchymal–epithelial interaction, which leads to the condensation of mesenchymal cells and differentiation into either a chondrogenic or osteogenic lineage [77]. When MSCs initially differentiate into chondrocytes and the subsequent cartilage is later resorbed and replaced by mineralized tissue, the process is known as endochondral ossification (ECO), which is responsible for the formation of long bones and most of the mammalian skeleton. When MSCs directly differentiate into osteoblasts at ossification centers, the process is known as intramembranous ossification (IMO), which is responsible for the formation of flat bones and the majority of the bones in the calvarial and orofacial region [77,78]. Angiogenesis and vascular ingrowth play a vital role in both ossifications. In IMO, capillaries penetrate into the differentiating mesenchymal zone, whereas in ECO, the hypertrophic chondrocytes recruit the infiltrating vasculature, leading to the arrival of osteoblasts, the resorption of hypertrophic cartilage, and the mineralization of the matrix [79].

Unlike natural bone development where ECO and IMO occur in different parts of the body and form different bone structures, bone fracture healing involves both ECO and IMO in the same region [80]. The bone fracture healing process starts with an initial acute inflammatory response, which attracts MSCs to the site. The mechanically less stable regions next to the fracture site are stabilized by the formation of cartilage via ECO, while the further stabilization of the fracture occurs concurrently through the formation of a hard callus across the fracture gap via IMO [81]. This primary bone production is followed by bone remodeling, in which the original bony callus is altered by resorption and subsequently by a secondary bone formation to restore the functional load-bearing anatomical structure [82].

To effectively employ AD-MSCs for bone regeneration, it is important to understand the roles of MSCs and the molecular mechanisms behind bone development and bone repair. Both ECO and IMO begins with the condensation of MSCs, involving cell adhesion molecules such as neural-cell adhesion molecules (N-CAMs) and neural cadherin (CDH2) [77]. Following condensation, the recruited MSCs differentiate into chondrocytes in ECO, or into osteoblasts in IMO. The chondrogenic or osteogenic differentiation of MSCs is a multifaceted process orchestrated by the interplay of various signaling pathways and regulatory molecules. In ECO, the factor SRY (Sex-determining region Y) box 9 (SOX9) serves as the key transcription factor in chondrocyte differentiation, as it regulates the expression of collagen II, collagen IX, collagen XI, aggrecan, and other cartilage matrix proteins [83,84]. In IMO, the initiation of osteogenic differentiation is marked by the upregulation of osteogenic genes including runt-related transcription factor 2 (RUNX2) and osterix (OSX). Simultaneously, there is a downregulation of genes associated with alternative lineages, such as adipogenic peroxisome proliferator-activated receptor-g (PPARγ) and CCAAT-enhancer-binding protein a (C/EBPa). This regulatory shift results in the expression of relevant proteins, including alkaline phosphatase (ALP), osteopontin (OPN), osteonectin (OCN), and the initiation of matrix mineralization [51].

Several key biological mediators, such as bone morphogenic proteins (BMPs), Indian Hedgehog (Ihh), parathyroid hormone related peptides (PTHrPs), vascular endothelial growth factor (VEGF), and fibroblastic growth factors (FGFs), serve vital roles in both ECO and IMO [85]. BMP is a member of the transforming growth factor-β (TGF-β) family and the 16 subtypes of BMPs have varying chondro-inductive and/or osteo-inductive activities [86]. For example, BMP-2 has been shown to induce the osteogenic differentiation of AD-MSCs while BMP-7 stimulates AD-MSCs into a chondrogenic phenotype [87]. In ECO, BMP-4 is important for early-stage MSC chondrogenic differentiation while BMP-6 mediates chondrocyte hypertrophy [88]. Ihh and PTHrP perform the critical function of regulating chondrocyte proliferation and hypertrophy in ECO and inducing uncommitted MSCs to pre-osteoblasts in IMO [83]. VEGF induces angiogenesis and is, therefore, vital in both ECO and IMO. FGFs, a large family of 22 proteins (in human or mice), perform diverse functions in vertebrates by binding to receptors that activate biological signaling pathways [83]. FGFs are involved in bone development from the earliest stage of bud development to the final phases of ossification, with FGF2 being observed in chondrocytes, osteoblasts, and periosteal cells. FGF7, FGF8, and FGF17 are expressed in the perichondrium during ECO, while FGF9 and FGF18 are shown to regulate bone growth plate development [86].

Two major osteogenic signaling pathways, Wnt/β-Catenin and BMP/Smads, play pivotal roles in AD-MSC osteogenesis. In these pathways, β-Catenin and Smads proteins in the nucleus interact with the transcription factors Runx2 and Osterix, influencing the transcription of osteogenesis-related genes and promoting the differentiation into osteoblasts [89]. The Wnt/β-Catenin pathway also plays a role in regulating chondrocyte differentiation in ECO, as Wnt proteins tend to inhibit chondrocyte differentiation in favor of osteogenic differentiation [77]. Mitogen-activated protein kinase (MAPK) signaling, specifically involving extracellular signal-regulated kinases (ERK), jun N-terminal kinases (JNK), and p38MAPK, significantly contributes to AD-MSC osteogenesis [90], while Notch signaling interacts with BMP and Wnt pathways to inhibit chondrocyte differentiation in ECO and modulate the osteogenic differentiation process in IMO [83,91].

Supplementation by soluble factors or external stimuli is essential for reaching the later stages of osteogenic differentiation with AD-MSCs. Dexamethasone, one of the most prevalent soluble factors, regulates osteogenic differentiation by activating Wnt/β-catenin and BMP signaling and can prompt osteogenesis via the phosphorylation of Runx2 by mitogen-activated protein kinase phosphatase 1 (MKP-1) [92]. It has concentration-dependent osteoinductive effects which balances both osteogenic and adipogenic differentiation [35]. Other factors such as retinoic acid and ascorbic acid also contribute to enhancing the osteogenic potential of ADMSCs. Retinoic acid has also been shown to increase the effect of BMP-2 on osteogenic differentiation of human ASCs [93]. Ascorbic acid is crucial for collagen synthesis, and in its absence, collagen chains fail to form the appropriate helix. Osteoblast interactions with the collagen-containing extracellular matrix trigger MAPK signaling, activating Runx2 phosphorylation and inducing osteoblast differentiation. β-Glycerophosphate provides phosphate crucial for hydroxyapatite formation, contributing to the mineralization process [92]. Knowing and understanding these intricate mechanisms is vital for optimizing protocols in regenerative medicine and tissue engineering applications.

### 2.5. Cell Delivery: Homogeneous AD-MSCs versus SVF

Traditionally, researchers using AD-MSCs in regenerative medicine isolate AD-MSCs from adipose tissue by enzymatic digestion using collagenase followed by centrifu-gation and cell selection via plastic adherence [94]. The plastic adhering characteristic of AD-MSCs allows researchers to selectively expand and obtain a homogeneous population of AD-MSCs, which can be used for autologous or allogeneic transplantation. Homo-geneous AD-MSCs are well-tolerated in allogeneic hosts [95] as MSCs lack the human leucocyte antigen class II molecules responsible for immune rejection [96].

Over the years, it has been increasingly common for AD-MSCs to be transplanted autologously in the form of SVF [97]. SVF can be categorized into two types based on its mode of acquisition: cellular SVF (cSVF), which is obtained by the enzymatic digestion of adipose tissue by collagenase, and tissue SVF (tSVF), which is obtained by the mechanical disruption of subcutaneous adipose tissue followed by aspiration with a syringe or vacuum pump [98]. Although SVF is unsuitable for allogeneic treatments due to the presence of various cell types causing an immune reaction, SVF has two main advantages over homogenous AD-MSCs for autologous treatments: (1) SVF is easier and faster to obtain than homogenous AD-MSCs as there is no need for cell culture and cell expansion, and (2) the heterogeneous cellular composition of SVF can contribute numerous benefits to the tissue regeneration process, including immunomodulation, anti-inflammatory responses, and angiogenesis [99].

Although the use of collagenase digestion to generate cSVF is highly effective and offer high yields (2 to 6 million cells per ml lipoaspirate with ≥90% cell vitality) [100], it is expensive, time-consuming, and poses challenges to regulatory compliance [101,102]. According to the “minimal manipulation” guidelines on autologous implantation set by regulatory agencies, adipose tissue meant for autologous implantation should be minimally manipulated, intended for homologous use, and not combined with other articles [103]. The use of collagenase is thus considered “more than minimally manipulated” as collagenase alters the original characteristics of adipose tissue, potentially affecting the phenotypical and functional characteristics of the isolated cells [94,104]. To overcome this, enzyme-free methods such as mechanical disruption can be employed to breakdown the extracellular matrix (ECM) of the adipose tissue and generate tSVF [105]. tSVF has a variable cell population similar to the cSVF, and also contains cellular debris, blood cells, and ECM fragments [106]. One advantage of using tSVF instead of cSVF or homogeneous AD-MSCs is that the native ECM and perivascular structures present in tSVF provide structural support for the AD-MSCs, helping to reduce cell death and improve graft retention [107]. The use of tSVF also eliminates the need for enzyme digestion and cell expansion, allowing for the extraction, processing, and implantation to be completed within a single surgery [108]. The isolation of tSVF also meets the “minimal manipulation” guidelines in autologous implantation.

### 2.6. Towards Xeno-Free Culture and Xeno-Free Osteogenic Differentiation of AD-MSCs

While the use of AD-MSCs in regenerative medicine has surged, their clinical application has been hampered by the presence of animal-derived components [95]. In order to seed AD-MSCs into a scaffold, in vitro expansion is necessary to obtain sufficient numbers for cell-based therapy. Traditional culture methods involving fetal bovine serum (FBS) or fetal calf serum (FCS) are ill-defined with batch-to-batch variations, prompting a shift to chemically defined serum- and xenogeneic-free culture media [109,110,111,112]. These transitions are vital due to the inherent variability of these animal-derived components and the associated risks, such as inter-species contamination and immunological reactions [113,114,115]. Consequently, numerous studies have been dedicated to eliminating animal-derived components to develop a fully defined serum-free media (SFM) for AD-MSCs expansion or differentiation in clinical use [109,116,117,118].

A comparative analysis by Lee et al. [119] highlights the benefits of SFM for human AD-MSCs, demonstrating lower cellular senescence, immunogenicity, and higher genetic stability than in FBS-containing media. Several studies demonstrate that hAD-MSCs expanded in various SFMs can maintain their stem cell phenotype and tri-lineage differentiation capability [109,118,119,120]. In another study, Al-Saqi et al. [120] compared the growth of hAD-MSCs in commercially available, defined serum-free GMP-grade media (SFM) and traditional DMEM media supplemented with FBS (DMEM-FBS). The results reveal that hAD-MSCs cultured in SFM maintained a stable morphology over five passages, had a shorter population doubling time than DMEM-FBS, and preserved their stem cell phenotype and tri-lineage differentiation potential. Interestingly, hAD-MSCs in SFM exhibited a higher osteogenic potential than in DMEM-FBS, consistent with the findings from other studies [121,122].

The successful osteogenic differentiation of hAD-MSCs using serum- and xenogeneic-free osteogenic differentiation media has been reported in multiple studies [111,123,124]. Ochiai et al. [111] demonstrated the positive osteogenic differentiation of hAD-MSCs cultured in a complete serum- and xeno-free protocol, using RoosterNourish^TM^-MSC-XF (RoosterBio, Frederick, MD, USA) for expansion and the OsteoMAX-XF Differentiation medium (Merck Millipore, Burlington, MA, USA) for differentiation, with positive alizarin red staining and osteocalcin expression. However, it is noteworthy that these commercial serum-free media effective for human AD-MSCs may not be suitable for other species. A study by Devireddy et al. [110] revealed that the commercial serum-free media, RoosterBio SF (RoosterBio, Frederick, MD, USA), did not support the growth of canine AD-MSCs, emphasizing the need to explore different formulations for expanding AD-MSCs derived from various species.

Overall, chemically defined serum-free media present a promising alternative to FBS-containing media, demonstrating significant potential for clinical cell-based applications by eliminating animal-derived components, mitigating potential risks, and reducing variability. Thus, the development of a fully serum- and xeno-free protocol for cell expansion and differentiation holds great promise in facilitating clinical translation and securing FDA approval for regenerative medicine applications.

### 2.7. AD-MSCs Secretome during Bone Regeneration

#### 2.7.1. The AD-MSC Secretome

Fracture healing is a complex process influenced by the crosstalk of surrounding tissues and related wound factors. AD-MSCs produce numerous molecules responsible for cell signaling, such as cytokines [125], growth factors [126], morphogens [127], chemokines [128], and extracellular vesicles [125], which improve various cellular mechanisms. The AD-MSCs’ secretome influences the surrounding tissue in multiple ways. It supports angiogenic and osteogenic differentiation, and progenitor cells are recruited to the fracture site, thereby promoting bone regeneration. Furthermore, AD-MSCs possess immunomodulatory properties [43], and it has been demonstrated that they interact with the innate immune system and reduce the number of B-cells at the fracture site, thus facilitating bone growth [129].

#### 2.7.2. Cytokines and Growth Factors

AD-MSCs are characterized by high trophic activity and the secretion of a large amount of proteins, growth factors, and pro- and anti-inflammatory cytokines, which exert benefits towards the cells’ regenerative capacity. They are considered highly immunomodulating cells, exceeding the suppressive effect of BM-MSCs by secreting more anti-inflammatory IL-6 and transforming growth factor-β1 (TGF-β1) [130]. Relevant levels of IL-2 affect AD-MSCs function by transcriptional dysregulation [131], while IL-6 enhances ALP activity, promotes osterix expression, and thus osteogenesis [132,133].

AD-MSCs secrete other cytokines with well-defined pro-inflammatory (IL-7, IL-8, IL-9, IL-11, IL-12, IL-15, IL-17, IFN-γ, IL-1β, and TNF-α) or anti-inflammatory (IL-1Ra, IL-4, IL-10, and IL-13) properties [134]. TNF-α and IL-1β secretion by macrophages mediates the inhibitory effect on ASCs adipogenesis, and the combination of TNF-α or IL-1β with IFN-γ can enhance the immunomodulatory properties of AD-MSCs, mainly dependent on indoleamine 2, 3-dioxidase (IDO) or inducible nitric oxide synthase (iNOS) [135]. IFN-γ triggers AD-MSCs to elicit immuno-suppressive factors [136], while the signaling protein PGE2 secreted by AD-MSCs has an immunosuppressive effect [137]. IL-4 and IL-17 have an inhibitory effect on adipogenic differentiation by promoting lipolysis and suppressing proadipogenic factors’ gene expression, respectively [138].

In addition to cytokines, AD-MSCs secrete many growth factors that influence cellular processes promoting regeneration. The proangiogenic and antiapoptotic properties of AD-MSCs are provided by trophic factors such as VEGF, FGF-2, TGF-β, HGF, and GM-CSF [9]. HGF and VEGF also induce neurogenic responses [139]. The secretion of HGF, involved in vasculogenesis and angiogenesis [140], is significantly increased after AD-MSCs stimulation with FGF-2 or EGF. PDGF secreted by AD-MSCs plays an essential role in angiogenesis [141], and cell stimulation with PDGF enhances the release of extracellular vesicles and thus proangiogenic properties [142]. AD-MSCs also secrete an insulin-like growth factor (IGF), promoting the proliferation, self-renewal, and differentiation of cells [143], but its level decreases with the donor age [144]. Furthermore, IGF-1, EGF, FGF-2, and TGFα are essential wound healing factors, enhancing these processes and cell migration [145].

#### 2.7.3. Extracellular Vesicles (EVs)

Extracellular vesicles (EVs) play a crucial role in tissue crosstalk and stem cell regulation. They can be found in nearly all tissues and have garnered significant attention in recent years. EVs are involved in the horizontal transfer of mRNA, miRNA, other non-coding RNAs, proteins, lipids [146], and mitochondrial DNA [147]. In the context of cell therapy and tissue engineering, two primary types of EVs are discussed: exosomes and microvesicles. Exosomes, typically ranging from 30–150 nm in diameter, originate from the endosomal pathway [148]. They bud from the endosomal membrane into the lumen, creating the multivesicular endosome. In contrast, microvesicles are larger, measuring 100–1000 nm, and are formed through an outward budding of the cellular membrane regulated by small GTPases [149]. EVs are released when the multivesicular endosome fuses with the cellular membrane [150]. EVs influence tissue crosstalk through surface markers, such as membrane molecules [150], and their cargo, including regulatory RNAs and signaling pathway factors [151]. It has been shown that EVs can act directly and bind to specific cells [150].

EVs play a significant role in bone regeneration, effectively promoting bone repair and regeneration independently, and exerting immunomodulatory effects by binding to receptors and promoting osteogenesis [152,153]. Current research suggests that adipose-derived mesenchymal stem cell-derived extracellular vesicles (AD-MSC-EVs) can promote osteogenic differentiation and angiogenesis, regulate immune function, induce chondrogenesis, and improve osteoporosis [154,155]. EVs facilitate a process commonly known as cell-free therapy, promoting cell proliferation, tissue regeneration, and neovascularization at the injured site by releasing a large number of miRNAs, cytokines, and other bioactive substances through AD-MSC-EVs. Unlike stem cells, which carry the risk of causing tumors after long-term culture, AD-MSC-EVs do not pose such a problem. Compared to AD-MSCs, AD-MSC-EVs exhibit lower immunogenicity, reduced chances of immune rejection, and are easier to preserve and transport [156].

EVs exert osteogenic effects by influencing the phosphatidylinositol 3-kinase/protein kinase B (PI3K/AKT) signaling pathway [157] and through miRNA196a [158]. Moreover, they are involved in the recruitment of mesenchymal stem cells (MSCs) toward a fracture site [159]. EVs have also been found to exhibit pro-angiogenic [158] and pro/anti-inflammatory effects [157]. In murine calvarial defects, AD-MSC-derived EVs delivered via poly glycolic-co-lactic acid/poly dopamine scaffolds resulted in significantly higher levels of new bone formation compared to non-EV scaffolds [160]. In a subcutaneous implantation model in nude mice, AD-MSC-EVs functionalized onto MG63 cell-seeded titanium substrates stimulated the greater osteogenic differentiation of the seeded cells compared to the controls [161]. Interestingly, while these EVs did not support the osteogenic differentiation of AD-MSCs in vitro, they expressed high levels of miR-21, let-7f, miR-10a&b, and miR-199b, all of which are involved in maintaining bone homeostasis or promoting osteogenic differentiation via SMAD, RUNX2, GSK3β/catenin, Axin2, and/or Krüppel-like factor (KLF)4 signaling [161]. Ma et al. discovered that by loading therapeutic mRNAs, including BMP-2 and VEGF-A mRNAs, together with associated microRNAs into EVs, they are capable of stimulating osteogenesis and angiogenesis in a challenging critical-size defect model in rats [162]. In another study, Li et al. [160] found that embedding EVs from AD-MSCs in a scaffold increased healing after six weeks in a mouse model with a calvarial defect.

Despite these multiple findings, the regulatory influence of EVs is not fully understood; it seems to depend on specific tissues and is highly dependent on other factors [163]. For example, Zhu et al. [164] found that type 1 diabetes reduces the osteogenic effects of EVs. Overall, EVs are involved in promoting bone regeneration in various ways, including the regulation of the immune environment, promotion of angiogenesis, differentiation of osteoblasts and osteoclasts, and promotion of bone mineralization. They contain fewer membrane proteins, making clinical applications safer and with higher yields. Furthermore, for cell-free therapy using AD-MSC-EVs, the main challenges are the high clearance rate, short half-life, and easy inactivation of this complex. The amounts of EVs produced and the active substances contained within them vary among different AD-MSCs and the same AD-MSCs under various physiological conditions, thus ensuring quality is also a challenge. To date, there is no efficient and accurate method for the isolation and detection of AD-MSC-EVs, and monitoring, controlling, and regulating their biological activity and function remain unresolved. Therefore, more studies should be conducted to understand this mechanism. Despite these challenges, EVs are expected to be an ideal component to combine with bone engineering scaffolds to guide bone regeneration [165].

## 3. AD-MSCs in Pre-Clinical Applications

### 3.1. In Vitro Applications

In vitro studies have consistently demonstrated the multipotency of AD-MSCs (Table 1) through their cellular mechanics, such as signaling pathways and cellular stimulation for the osteogenesis potential [43]. The culture techniques and methods of analysis of AD-MSCs usually involve the harvesting, isolation, culture, identification, and analysis of the secretory profiles in stem cell proliferation and osteogenic differentiation. The capacities of the proliferation and osteogenic differentiation of the stem cells are often investigated through metabolic assays such as the 3-(4,5-dimethylthiazol-2-yl)-2,5-diphenyltetrazolium bromide (MTT) assay and the gene expression analysis of common osteogenic markers, alkaline phosphatase activity (ALP), and genes related to calcium mineral deposition and bone formation. The adherent capability of the cells is critical for cells to proliferate, thus the adhesion potential of the cells is observed through scanning electron microscopy (SEM) and micro-computed tomography. Overall, in vitro studies aim to assess the growth and adhesion of cells, aimed at demonstrating AD-MSCs’ multipotency through inducing appropriate differentiation stimuli, followed by lineage-specific identification and gene expression profiling, to verify their feasibility in supporting the osteogenesis process.

**Table 1 ijms-25-06805-t001:** Selected in vitro studies of AD-MSCs for bone tissue engineering.

First Author	Cell Source and Type	AD-MSCs’ Harvest Method	Scaffold Used	Key Findings
Sari et al. [166]	AD-MSCs from rat	Not Reported	Bovine teeth scaffold	AD-MSCs exhibited good cell adherence to the allograft, showing its biocompatibility and accelerated osteogenic differentiation of AD-MSCs.
Kurzyk et al. [167]	AD-MSCs from human	Lipoaspirate	PCL PCL + 5% TCP	AD-MSCs exhibited stable proliferative capacity and can be cultured for long durations in vitro. AD-MSCs were biocompatible with PCL or PCL + 5%TCP scaffolds, but PCL + 5%TCP scaffold showed the best bone regenerative capacity.
Hosseini et al. [168]	AD-MSCs, BM-MSCs, and USSCs from human	Buccal Fat Pad	Bioceramic PCL PCL-Bio	Proliferation rate between 3 stem cell types was not significantly different. Higher proliferation was evident on scaffold compared to tissue culture place. BFP-AD-MSCs was concluded to be the best cell source type due to its availability and easy harvesting method.
Gandolfi et al. [169]	AD-MSCs from human	Purchased	Mineral-doped PLA-based porous scaffolds	Mineral-doped scaffolds showed a dynamic surface and created a suitable bone-forming microenvironment—increasing osteogenic commitment. Presence of exosomes increased osteogenic gene markers. Exosome-enriched scaffolds could improve bone regenerative capacity.
Roato et al. [170]	SVF-AD-MSCs from human	Lipoaspirate	Xenohybrid bone scaffold	SVF-AD-MSCs with scaffold, in the presence of osteogenic factors, had higher osteoinductive capabilities than AD-MSCs.
Ghaderi et al. [171]	AD-MSCs and GDCs from human	Buccal Fat Pad	No Scaffold	Both BFP-AD-MSCs and GDC demonstrated potential to differentiate into osteocyte and chondrocyte, but BFP yielded a greater proportion.
Ahmed et al. [172]	AD-MSCs and BM-MSCs from human	Lipoaspirate	No Scaffold	AD-MSCs showed significantly higher proliferation and adipogenic capacity with more lipid vesicle formation and expression of the adipogenesis-related genes than BM-MSCs. In contrast, BM-MSCs showed significantly higher osteogenic and chondrogenic capacity compared to AD-MSCs.
Mazzoni et al. [173]	AD-MSCs from human	Lipoaspirate	HA	Scaffold provided the ideal microenvironment for AD-MSCs adhesion, increasing proliferation and upregulation of osteogenic genes with improvement in matrix mineralization and cell viability.
D’Alimonte et al. [174]	AD-MSCs and DPSCs from human	Purchased	Titanium disks	AD-MSCs had greater proliferation, osteogenic differentiation, and colony-forming ability than DPSCs. AD-MSCs also showed better colonization and adhesion on the titanium scaffold.
Canciani et al. [175]	AD-MSCs from human	Subcutaneous	30/70 HA b-TCP	Combination of AD-MSCs and HA/b-TCP scaffold increased alkaline phosphatase activity of the cell and cellular vitality increased. Good adhesion capacity was observed between the cell and scaffold interface.
Calabrese et al. [176]	AD-MSCs from human	Adipose tissue biopsies/lipoaspirate	Collagen/Mg doped HA	Collagen/hydroxyapatite scaffold can induce AD-MSCs differentiation and the addition of osteoinductive factors accelerates the osteogenic process.
Russell et al. [39]	BM-MSCs and AD-MSCs from canine	Subcutaneous	No Scaffold	No significant differences were found between cell types in terms of their adipogenesis, osteogenesis, immunomodulatory capacity, immunophenotype, and progenitor and non-progenitor functions. AD-MSCs had higher isolation success and proliferation rate.
Caetano et al. [177]	AD-MSCs from human	Lipoaspirates	PCL	AD-MSCs were viable in the scaffolds, and they could differentiate toward the osteogenic lineages for 28 days in culture with osteogenic medium.
A. Ardeshirylajimi et al. [178]	BM-MSCs, AD-MSCs, and USSCs from human	Buccal Fat Pad Lipoaspirate	PLLA PLLA-Bio TCPS	No significant difference between proliferation levels of the four types of stem cell. BFP-AD-MSCs exhibited osteogenic differentiation close to BM-MSCs. Bio-Oss-coated PLLA found to be most appropriate substrate to support proliferation and osteogenic differentiation of stem cells in vitro.
Kishimoto et al. [179]	AD-MSCs from human	Buccal Fat Pad	No Scaffold	DFAT cells’ osteoblastic differentiation ability is higher than that of ASCs. We consider DFAT cells from the BFP to be an ideal cell source for bone tissue engineering.
Requicha et al. [180]	AD-MSCsfrom canine	Abdominal tissue	Starch and PCL	Wet-spun fiber mesh layer functionalized with silanol groups stimulated the osteogenic differentiation of AD-MSCs, while the membrane layers enabled a good cell attachment and proliferation. Double-layer scaffold enhances osteogenesis and promotes colonization.
Niada et al. [181]	AD-MSCs from pigs	Subcutane-ous Buccal fat pad	Titanium disks	AD-MSCs from BFP osteo-differentiated well in association with synthetic supports. BFP contains a population of progenitor cells with stemness features that can differentiate in vitro and are associated with synthetic supports.
Melief et al. [182]	AD-MSCs and BM-MSCs from human	Cadaveric Pancreata	No Scaffold	Immunomodulatory capacities of BM-MSCs and AD-MSCs are similar, AD-MSCs have more potent immunomodulatory effects than BM-MSCs where lower numbers of AD-MSCs evoke the same level of immunomodulation.
Broccaioli et al. [183]	AD-MSCs from human	Buccal Fat Pad Subcutaneous	Alveolar bone Periodontal ligament Collagen membrane	Both AD-MSCs harvested from BFP and subcutaneous region have good proliferation rate and adhered to bone, periodontal ligament, collagen membrane, and polyglycol acid filaments. BFP-AD-MSCs were found to be able to differentiate and adhere to biological supports and synthetic materials. They are also able to proliferate in the presence of human serum.
Jia Liu et al. [184]	AD-MSCs from rat	Subcutaneous	Heterogeneous deproteinized bone (HDB)	AD-MSCs-HDB composite displayed strong osteogenic ability, able to regenerate bone for segmental bone defects. It has significant improvements compared to HDB only method, showing higher density in the AD-MSCs experimental groups
Shi et al. [185]	AD-MSCs (SVF) from human	Lipoaspirate	Human cancellous bone	AD-MSCs derived from the SVF of adipose have all the characteristics of MSCs, which include adherence, the presence of CD markers, and the capability of tri-lineage differentiation.
PEÑA et al. [186]	AD-MSCs and BM-MSCs from human	Lipoaspirate	No Scaffold	AD-MSCs and BM-MSCs display distinct immunophenotypes based on surface positivity and expression intensity as well as differences in adipogenic differentiation.
Guasch et al. [187]	AD-MSCs from human	Lipoaspirate Buccal Fat Pad	No Scaffold	AD-MSCs from BFP differentiate to chondrocytes, osteoblasts, and adipocytes, suggesting that BFP can be a rich alternative source of stem cells.
Gabbay et al. [188]	AD-MSCs from human	Lipoaspirate	No Scaffold	Greater expression of osteogenic markers in AD-MSCs were shown in the 3-dimensional collagen gel—cells were found to adhere more readily to the 3-dimensional structure.
Halvorsen et al. [36]	AD-MSCs from human	Not Reported	No Scaffold	Adipose tissue-derived human stromal cells can be expanded more than 100-fold, displaying adipocyte-specific proteins and osteoblastic gene markers. A readily available source of multipotential stromal cells.
Ugarte et al. [189]	AD-MSCs and BM-MSCs from human	Lipoaspirate	No Scaffold	No significant differences were observed for yield of adherent stromal cells, growth kinetics, cell senescence, multi-lineage differentiation capacity, and gene transduction efficiency. Adipose tissue is an abundant and easily procured source of PLA cells, applicable for tissue-engineering and as gene delivery vehicles.
Zuk et al. [31]	AD-MSCs from human	Lipoaspirate	No Scaffold	Adipose tissue may be another source of pluripotent stem cells with multi-germline potential.

#### 3.1.1. AD-MSCs as a Cell Source for Bone Regeneration

Studies comparing AD-MSCs with BM-MSCs have shown that in vitro osteogenic responses of AD-MSCs were comparable to that of BM-MSCs displaying a common expression profile for many surface antigens. While AD-MSCs showed a higher adipogenic differentiation capability than BM-MSCs [172,186], the higher isolation success and proliferation rate and less tedious culturing process of AD-MSCs make AD-MSCs more attractive than BM-MSCs for osteogenic applications.

A study by Samih et al. [172] comparing the proliferation of human AD-MSCs and BM-MSCs up to 21 days demonstrated that AD-MSCs had higher proliferation (Figure 1). In addition, immunomodulatory capacities were higher in AD-MSCs than those of BM-MSCs, exhibiting the increased secretion of cytokines that have implicated the immunomodulatory modes of action of multipotent stromal cells, such as interleukin-6 and transforming growth factor. This is correlated with the higher metabolic activity of AD-MSCs compared to BM-MSCs, suggesting that AD-MSCs could be considered as a good alternative for immunomodulatory therapy [182].

**Figure 1 ijms-25-06805-f001:**
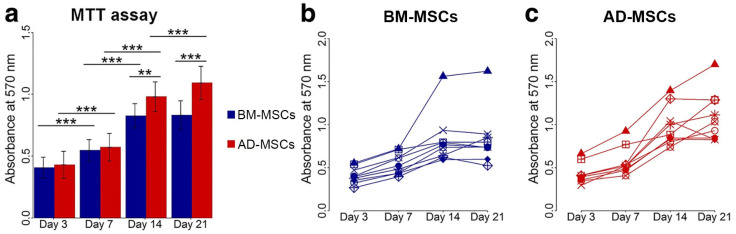
Proliferation of AD-MSCs and BM-MSCs across 4 different time points. (**a**) MTT assays showed higher cellular metabolic activity in AD-MSCs compared to BM-MSCs and higher number of cells were present at day 14 and 21 (Average of 9 donors). ** *p* < 0.01, *** *p* < 0.001. (**b**,**c**) Proliferation of BM-MSCs and AD-MSCs. MTT expression in each donor. Each symbol represents one donor. Reproduced with permission from Mohamed-Ahmed et al. Adipose-derived and bone marrow mesenchymal stem cells: a donor-matched comparison. Stem Cell Research & Therapy 2018, 9, 168, Figure 2 (https://doi.org/10.1186/s13287-018-0914-1) under the terms and conditions of the Creative Commons Attribution (CC BY) license (http://creativecommons.org/licenses/by/4.0/ accessed on 14 October 2023) [172].

Comparing other cell lines with AD-MSCs in vitro, AD-MSCs significantly stood out, showing positive outcomes in stimulating bone regeneration. The proliferation rates of different cell types—BM-MSCs, Unrestricted somatic stem cells (USSCs), and Buccal Fat Pad (BFP)-derived AD-MSCs—were not significantly different to each other when cultured on PCL-Bio, but they exhibited superiority in specific factors (Figure 2). BM-MSCs achieved the highest mineralization, USSCs showed higher gene expression, and BFP-AD-MSCs demonstrated the highest ALP activity. Different types of stem cell sources exhibited different abilities, but overall, BFP-AD-MSCs had a better standing in comparison to other cell lines due to its availability, less tedious preparation procedure, and reduced patient suffering [168].

**Figure 2 ijms-25-06805-f002:**
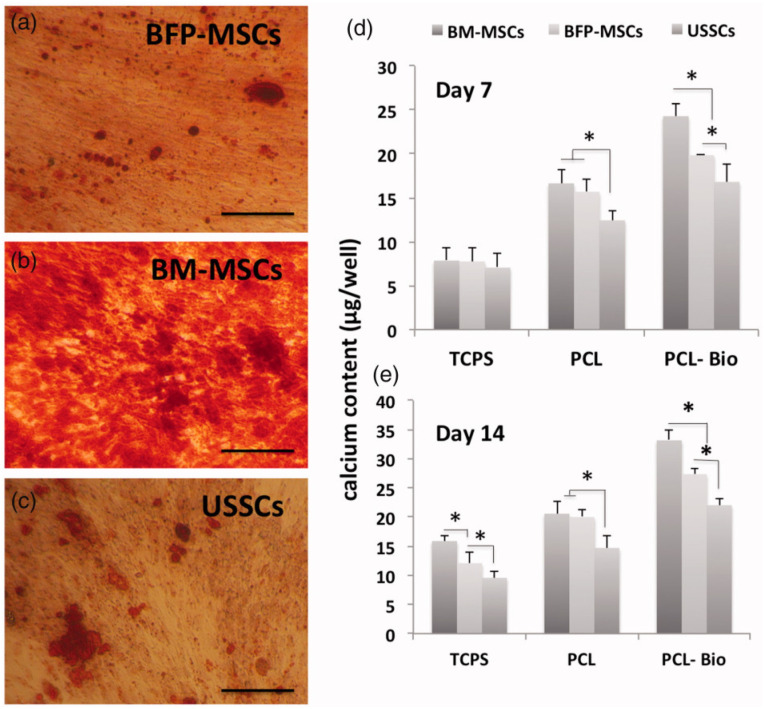
Alizarin red staining of different cell indicating calcium-containing osteocytes. All cells were stained after 2 weeks of culture on tissue culture polystyrene. (**a**) AD-MSCs from buccal fat pad (BFP-MSCs), (**b**) BM-MSCs, (**c**) unrestricted somatic stem cells (USSCs). Calcium content present on different scaffolds—TCPS, PCL, and PCL-Bio for (**d**) 7 days and (**e**) 14 days, respectively. * *p* < 0.05. Reproduced with permission from Hosseini et al. In vitro osteogenic differentiation of stem cells with different sources on composite scaffold containing natural bioceramic and polycaprolactone. Artif Cells Nanomed Biotechnol 2019, 47, 300–307, Figure 5 (https://doi.org/10.1080/21691401.2018.1553785) under the terms and conditions of the Creative Commons Attribution (CC BY) license (http://creativecommons.org/licenses/by/4.0/ accessed on 11 October 2023) [168].

One of the major limitations of using AD-MSCs for bone tissue engineering is the necessity to expand them through in vitro culturing. This process requires tedious steps requiring both isolation and expansion steps along with restrictive regulatory clearance. SVF is a non-cultured fraction of AD-MSCs with a heterogenous cell population obtained from lipoaspirates; the use of SVF has thus garnered much interest due to the simplification of the isolation steps. When AD-MSCs derived from SVF were seeded onto demineralized human cancellous bone grafts, cell numbers and CD markers (CD105, CD90, and CD73), regardless of donor age, gender, and body mass index, were found to be consistent and showed a tri-lineage differentiation capability. AD-MSCs showed significant adherence to the bone matrix, permitting osteogenic activity [185]. AD-MSCs-SVF seeded on a commercially available xenograft (SmartBone) had a better osteoinductive capability that surpassed the expanded cultured AD-MSCs. Though SVF is a heterogenous cell solution, its presence simulated mesenchymal cell activity better than AD-MSCs alone. Interestingly, even though osteogenic ALP-positive osteoblasts were present in SVF, they did not mineralize when left on their own. Mineralization only occurred in the presence of a scaffold, confirming that the microenvironment present in the scaffold plays an important role in stimulating the activation of these cells [170]. The heterogenous cell properties of SVF together with the scaffold environment thus appears to stimulate mesenchymal cell activity better than AD-MSCs alone.

#### 3.1.2. AD-MSCs from the Oral Region for Oral–Maxillofacial Applications

AD-MSCs for clinical applications are usually harvested from subcutaneous adipose depots, such as the abdomen, hips, and thighs. Given the abundance of adipose tissue in the body, researchers have been investigating the osteogenic differential capabilities of AD-MSCs from differing harvesting sites. For dental surgeons, there is a strong interest in harvesting AD-MSCs from the oral region, especially the buccal fat pad (BFP), for oral–maxillofacial bone regeneration as AD-MSCs from the BFP offer advantages such as easy accessibility from an intraoral source, a similar phenotype to AD-MSCs from abdominal adipose tissue, and the ability to differentiate into osteogenic cells [48,181]. More importantly, the harvest of BFP is a minimally invasive procedure with very low morbidity rates and can be carried out in a dental clinic with local anesthesia, unlike the harvest of adipose tissue from another part of the body which often requires full body general anesthesia [190].

Broccaioli et al. [183] compared AD-MSCs from the BFP with AD-MSCs from the subcutaneous region and found that both showed a high proliferation rate with good adherence to the surrounding bone as well (Figure 3). Additionally, AD-MSCs derived from BFP contains progenitor cells that osteo-differentiate well in association with synthetic supports [181]. Guash et al. [191] further demonstrated that the SVF isolated from BFP contained ~30% AD-MSCs with a high expression of the angiogenic marker CD34, suggesting that BFP can be an ideal source of stem cells for clinical applications. The osteoblastic differentiation ability was higher and harvesting methods were less tedious and did not incur an aesthetic impact. Overall, AD-MSCs harvested from BFP showed a population of stem cells that shared a similar phenotype with AD-MSCs from abdominal subcutaneous fat tissue, able to differentiate into chondrogenic, adipogenic, and osteogenic lineage [179].

**Figure 3 ijms-25-06805-f003:**
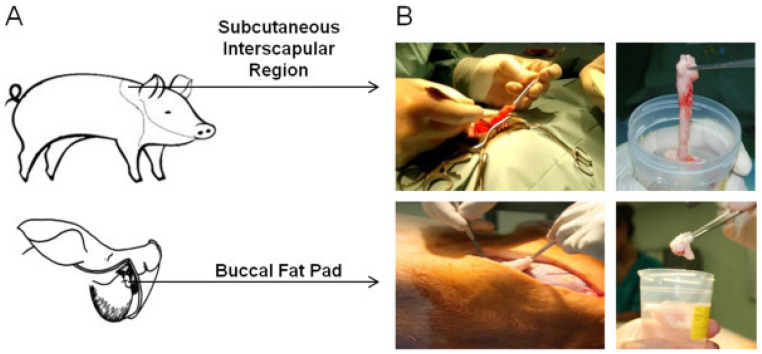
Different regions with the presence of AD-MSCs. (**A**) Anatomical regions of the pig for the harvest of AD-MSCs—subcutaneous and buccal fat pad tissue. (**B**) Surgical procedure for tissue collection. Reproduced with permission from Niada et al., A.T. Porcine adipose-derived stem cells from buccal fat pad and subcutaneous adipose tissue for future preclinical studies in oral surgery. Stem Cell Res Ther 2013, 4, 148, Figure 1 (https://doi.org/10.1186/scrt359) under the terms and conditions of the Creative Commons Attribution (CC BY) license (http://creativecommons.org/licenses/by/2.0/ accessed on 11 October 2023) [181] CC BY.

Gingiva soft tissues were compared to AD-MSCs derived from BFP. Though both cell types exhibited a good differentiation ability to osteocyte and chondrocyte, BFP-AD-MSCs yielded a greater proportion of stem cells [171]. Dental pulp stem cells were another alternative investigated and showed a similar fibroblast-like morphology and mesenchymal marker expression as AD-MSCs; however, doubling the time, coupled to a greater proliferation and colony forming ability, was found in AD-MSCs. When introduced onto titanium scaffold, AD-MSCs showed better colonization, resulting in higher osteogenic differentiation [174].

#### 3.1.3. AD-MSCs and the Role of Scaffolds

Scaffolds mimic the extracellular matrix by guiding the AD-MSCs to integrate well into the defect site, promoting the differentiation of progenitor cells while providing mechanical support. Such scaffolds should be biocompatible, eliciting minimal inflammatory and immunological responses when seeded with AD-MSCs. The biomaterial of the scaffold is a critical factor that drives osteogenesis processes. Scaffolds made with osteoconductive materials are efficacious with mainly partly differentiated cells (e.g., osteoblast and pre-osteoblast), but do not induce the osteogenic differentiation of osseous progenitor cells and AD-MSCs [192]. Meanwhile, scaffolds made with osteoinductive biomaterials demonstrate potential in recruiting progenitor cells and stimulating osteoblastic commitment and differentiation, promoting bone formation [193,194]. The different materials used to fabricate scaffolds range from human-derived allografts such as bovine teeth [166] (Figure 4), alveolar bone [183], heterogenous deproteinized bone (HDB) [184], bovine bone [195], cortical bone [196], and cancellous bone [185,197,198,199,200] to synthetic materials such as polycaprolactone (PCL) [167,177,180,201,202], Bioceramic [168], Polylactic acid (PLA) [169,178], hydroxyapatite (HA) [176,203,204], coral-hydroxyapatite (CHA) [205], strontium hydroxyapatite (srHA) [206], titanium [174,207,208,209,210,211], beta-tricalcium phosphate (bTCP) [175,187,212], poly(lactide-co-glycolide) acid (PLGA) [40,160,213,214], and fibrin gel [215,216].

**Figure 4 ijms-25-06805-f004:**
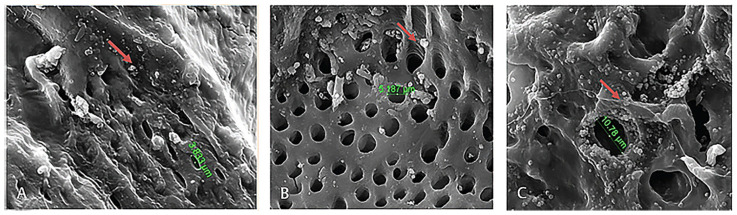
Adhesion of AD-MSCs on Bovine teeth scaffold demonstrated at different time intervals: (**A**) 1 h, (**B**) 12 h, and (**C**) 24 h. Arrow indicating the attachment point of AD-MSCs. The red arrow marked region is the attachment point. Reproduced with permission from Sari et al., Osteogenic Differentiation and Biocompatibility of Bovine Teeth Scaffold with Rat Adipose-derived Mesenchymal Stem Cells. Eur J Dent 2019, 13, 206–212, Figure 8 (https://doi.org/10.1055/s-0039-1694305) under the terms and conditions of the Creative Commons Attribution (CC BY) license (http://creativecommons.org/licenses/by/4.0/ accessed on 15 September 2023) [166].

In vitro studies have shown that scaffolds of different materials, when seeded with AD-MSCs, have a better osteogenic potential [166]. The adhesion of AD-MSCs onto the scaffold is a crucial parameter for proliferation to occur, in which the attachment and differentiation of cells were proportional—where more cells were found after 24 h in culture (Figure 3).

Liu et al. conducted a study to access the quality of bone formation by seeding AD-MSCs on heterogenous deproteinized bone (HDB). Four different models were introduced—(1) an osteogenic AD-MSCs group seeded on HDB, (2) AD-MSCs seeded on HDB, (3) non-seeded HDB, and (4) an untouched defect. The osteogenic AD-MSCs group combined with HDB demonstrated a higher callus density, showing a better rate and quality of new bone formation. Eight-week post-operative outcomes showed well-connected bone formation and the presence of a marrow cavity, with few defects identified, while the empty scaffold group had notable bony defects, although some degree of bone formation was present.

AD-MSCs were found to be viable with PCL and exhibited differentiation towards osteogenic lineages for 28 days in culture with the osteogenic medium [177]. However, when coated with TCP, PCL + 5% TCP had a better cell migration, adhesion, and infusion of nutrients, prompting the proliferation and differentiation of AD-MSCs for osteogenesis. The combination of PCL and TCP further enhanced bone regeneration compared to PCL on its own [167]. Modifying the scaffolds through coating and doping also enhanced osteogenic differentiation. Gandolfi et al. [169] demonstrated that mineral-doped PLA scaffolds showed a dynamic surface and created a suitable bone-forming microenvironment, where exosomes were easily entrapped on the surface of the scaffold that improved the gene expression of major markers of osteogenesis such as collagen type 1, osteopontin, osteonectin, and osteocalcin. This resulted in a higher osteogenic commitment of AD-MSCs.

Collagen and HA closely mimic both the cartilage and bone tissue [217,218]. For this reason, they have been frequently adopted as the base materials for scaffolds. Mazzoni et al. [173] cultured AD-MSCs on a HA-derived scaffold and AD-MSCs had a significant overall effect on the viability, demonstrating biocompatibility in terms of cell adhesion, morphology, and proliferation. Canciani et al. [175] also found increased alkaline phosphatase activity of AD-MSCs when seeded onto scaffolds composed of 30/70 HA and bTCP. Regardless of the culture time and conditions, the cells maintained their morphology and displayed adherence to the scaffolds, filling the macropores. bTCP enhanced the osteogenic potential of AD-MSCs to higher than that of undifferentiated cells. This suggests that the establishment of an enhanced gene expression pattern has a part to play in the differentiation of the cells grown on the scaffold and in the osteogenic medium [219]. Calabrese et al. [176] demonstrated that on a collagen/HA scaffold, AD-MSCs have the capability to differentiate into mature osteoblasts even in the absence of specific osteogenesis-inducing factors. Collagen/HA on their own were able to facilitate AD-MSCs’ differentiation, while the addition of osteoinductive factors accelerated the osteogenic process. Physiologically, collagen being the main component of bone tissue stimulates MSCs to differentiate into osteoblasts, initiating new bone formation [220]. Thus, the use of collagen/HA scaffolds seeded with AD-MSCs shows better proliferation and osteo-differentiation capacity as it closely mimics the natural bone regeneration process. Such scaffolds significantly enhance the osteogenic potential of seeded AD-MSCs cells.

The structural characteristics present in scaffolds provide binding sites for the cells and space for calcium deposits. The choice of the pore size and stiffness of the structure would determine the migration of cells and the vascularization outcomes, playing an important role for bone tissue formation. Modifying the architecture of the scaffolds has shown significant implications on osteogenic differentiation. Gabbay et al. [188] evaluated the effect of the extracellular matrix structure by seeding AD-MSCs in 2-dimensional (2D) monolayer configuration and 3-dimensional (3D) collagen gel cell culture conditions. A progressive stimulatory effect of AD-MSCs with regards to osteogenesis was found in the 3D gel compared to the 2D monolayer, with greater expressions of osteogenic markers. Requicha et al. [180] also studied the in vitro functionality of a double-layer scaffold. A double-layer fiber mesh layer functionalized with silanol groups stimulated the osteogenic differentiation of AD-MSCs, while the membrane layers enabled good cell attachment and a proliferation space. In such strategies, the double-layer scaffold proved to enhance osteogenesis, significantly prompting the colonization of cells with a distinct cellular population for bone regeneration. The use of unique architectures—specifically 3D structures—enhances the osteogenic differentiation of AD-MSCs.

Overall, the basic requirement for a tissue-engineered graft is the ability for seeded cells to adhere and proliferate, while also providing structural support and a bearing capacity. In vitro studies have shown that a scaffold’s materials and its architecture not only support structural integrity for bone regeneration but enhance the proliferation of AD-MSCs, accelerating osteogenic processes. The presence of growth factors, transplantation metabolites, degradation products, and the secretion of bioactive substances contribute significantly to the osteogenesis process and can be administered to expedite bone growth [221].

### 3.2. In Vivo Applications

Cells, scaffolds, and cytokines are key factors involved in osteogenesis and, to date, pre-clinical animal models have sought to investigate the feasibility and efficacy of scaffold implantation, cell implantation, or combined scaffold–cell implantations. The presence of different biological factors and the degree of tropism associated with the differentiation medium affects the osteogenic potential of AD-MSCs [139]. The use of in vivo models permits a more generalizable translational approach to identify the efficacy of AD-MSCs and their ability to regenerate bone. Additionally, a more detailed evaluation of the cell-to-scaffold-to-body interaction can be evaluated in pre-clinical animal models. Therefore, the development of in vivo models for bone tissue engineering is crucial in the translation of methods to clinical applications (Table 2).

#### 3.2.1. AD-MSCs’ Response in Bone Regeneration

The study of AD-MSCs for bone regeneration largely involves the response of AD-MSCs-scaffold implanted on defect sites of animal models prior to conducting a human clinical trial.

Cowan et al. [40] compared AD-MSCs and BM-MSCs seeded into HA-coated PLGA scaffolds for the recovery of a critical-size mouse calvarial defect. Both cell types demonstrated new bone formation at the 12-week time point, filling 70–90% of the area of the defect. AD-MSCs had a higher proliferation rate pre-implantation during the in vitro expansion and did not require any genetic manipulation nor the addition of exogenous growth factors.

Lee et al. [201] also compared AD-MSCs with BM-MSCs seeded on a PCL/TCP scaffold for the management of the maxillary bone defects of beagles. New bone formation was observed in both test groups. AD-MSCs treatment was concluded to be more superior than BM-MSCs due to its easy harvesting method and high cell yield.

Han et al. [216] showed a differing outcome when comparing BM-MSCs with AD-MSCs for the treatment of a large calvarial defect (10 mm × 10 mm), showing that BM-MSCs had higher differentiation into osteoblasts. However, the amount of bone regenerated at the end of 6 weeks showed no significant difference when comparing the two groups, indicating that there is a gap in the time for direct bone regeneration for AD-MSCs. Regardless, the final regeneration outcomes were identical due to indirect bone regeneration caused by the activation of bone regeneration proteins present in AD-MSCs.

When compared with cell types other than BM-MSCs, AD-MSCs consistently displayed better regenerative factors. Jin et al. [222] did a comparative characterization of MSCs harvested from dental pulp (DPSCs) and AD-MSCs of human subjects for the treatment of a 2 mm × 1 mm mandibular defect in 15 rats. DPSCs were found to have an enhanced colony-forming ability, higher proliferative ability, stronger migration ability, higher expression of angiogenesis-related genes, and secreted more vascular endothelial growth factor in comparison to AD-MSCs. AD-MSCs grew more slowly compared to DPSCs, but exhibited greater osteogenic differentiation potential, a higher expression of osteoblast market genes, and greater mineral deposition. Additionally, when implanted into the rats, AD-MSCs showed visible bone tissue formation in week 1 and exhibited a faster and greater bone regeneration capacity compared to the DPSCs. This indicates that the faster osteogenesis outcomes obtained in the in vitro studies do not necessarily reflect the bone regeneration capability in vivo.

**Table 2 ijms-25-06805-t002:** Selected in vivo studies of AD-MSCs for bone tissue engineering, grouped according to model.

First Author	Cell Source and Type	Harvest Method	Scaffold Used	Animal	Model	Key Findings
Ahn et al. [223]	AD-MSCs from human	Lipoaspirate	No Scaffold	Rat	8 mm calvarial defect	Decellularized AD-MSCs matrix loaded with bone morphogenetic protein BMP2 had effective bone regeneration without any immune side effects.
Dziedzic et al. [197]	AD-MSCs from rats	Inguinal fat	Decellularized Human Amniotic Membrane (DAM)	Rat	8 mm calvarial defect	DAM with AD-MSCs demonstrated higher host bone deposition and has shown to be effective in critical bone defect management.
Wang et al. [224]	AD-MSCs from rabbits EPCs	Inguinal adipose tissue	ADSC sheets	Rat and Rabbit	10 mm calvarial defects	ADSC sheet-EPC was osteogenic and EPC enabled the formation of capillary-like structures. The combined scaffold formed dense and well-vascularized new bone tissue at 8 weeks after implantation without any complications.
Maglione et al. [195]	AD-MSCs from rabbits	A single withdrawal from rabbits	Deproteinized bovine bone Bovine cancellous granular + collagen	Rabbit	2 mm, 6 mm calvarial defect	New bone was formed in the seeded scaffold and was similar to those obtained through traditional regenerative technique. AD-MSCs combined with scaffolds accelerated some steps in normal osseous regeneration.
Zhang et al. [225]	ADMCs from human	Not Reported	Osteogenic extracellular matrix (ECM) Small intestinal submucosa (SIS)	Rat	4 mm calvarial defect	AD-MSCs adhered faster and had better colonization on ECM-SIS scaffolds than on SIS scaffolds. Proliferation of AD-MSCs was promoted by the scaffolds without requiring additional osteogenic factors.
Semyari et al. [226]	AD-MSCs from rabbit	Fatty tissue from the nape	PLA PLAGA Decellularized amniotic membrane	Rabbit	8 mm circular calvarial defects	The scaffolds seeded with AD-MSCs showed development of well-vascularized bone tissues. AD-MSCs were osteoinductive, biocompatible, and promoted faster and more effective osteogenesis together with all types of scaffolds.
Ko et al. [227]	AD-MSCs from human	Purchased	Nanostructured decellularized tendon	Rat	4 mm calvarial defect	Nanostructured scaffolds had advantages over microstructure scaffolds as it enhanced cellular alignment, improving differentiation and regenerative potential of AD-MSCs resulting in accelerated bone regeneration
Di Bella et al. [228]	AD-MSCs	Inguinal fat pad	PLA	Rabbit	15 mm in diameter calvarial defect	PLA coated with fibronectin displayed significantly more bone formation within the scaffold matrix compared to non-coated group. The surface treatment of scaffolds with fibronectin enhances bone regeneration, due to the hydrophilic nature of fibronectin that permits greater cell adhesion, proliferation, and differentiation into the scaffold.
Han et al. [216]	AD-MSCs and BM-MSCs	Abdomen	Fibrin Glue	Rabbit	10 × 10 mm calvarial defect	AD-MSCs differentiate directly into osteoblasts less often than BM-MSCs. However, the total amount of regenerated bone is almost the same because of the effect of indirect bone regeneration.
Yoon et al. [214]	AD-MSCs from human	Lipoaspirate	PLGA	Rat	An 8 mm circular calvarial defect	Differentiated AD-MSCs combined with PLGA exhibited better, more robust bone regeneration capacity compared to undifferentiated AD-MSCs. Fourteen days of AD-MSCs culture duration was found to be optimal for differentiated AD-MSCs.
Probst et al. [213]	AD-MSCs from minipigs	Lower abdominal area	TCP-PLGA titanium osteosynthesis plates.	Mini Pigs	Mandibular defect	AD-MSCs-seeded scaffolds had higher osteocalcin deposition and newly formed bone in the defect area. Improved bone regeneration in large mandibular defects.
Jin et al. [222]	AD-MSCs and DPSCsfrom human	Lipoaspirate	No scaffold	Rat	2 mm mandibular bone defect, 1 mm thickness	AD-MSCs showed visible bone tissue as early as week 1 and promoted faster and greater bone regeneration (higher osteogenic differentiation potential, higher expression of osteoblast marker genes) compared to the DPSC group.
Mehra-bani et al. [199]	AD-MSCs from rabbits	Subcutaneous	Autologous bone graft	Rabbits	Bilateral 1.5 × 0.5 cm mandible defect	Significant increase in the thickness of new cortical bone when fibrin glue scaffold associated with AD-MSCs was used.
Lee et al. [201]	BM-MSCs and AMSCs from canine	Abdominal cavity	PCL TCP	Canine	Maxillary bone defect	AD-MSCs and BM-MSCs seeded onto 3D-printed PCL/TCP scaffolds are implanted in bone defects and showed similar osteogenic properties.
Pourebrahim et al. [204]	AD-MSCs from canine	Subcutaneous	HA bTCP	Canine	15 mm alveolar crest to the nasal floor defect	Bone formation with AD-MSCs were slower than that of autografts, but the rate increased rapidly after day 60, exhibiting comparable bone regeneration capability as autografts. AD-MSCs together with HA and bTCP showed good potency for bone regeneration in critical defects.
Lau et al. [108]	AD-MSCs	Lipoaspirate	PCL-TCP	Pigs	4 defect 8 mm × 8 mm in alveolar ridge	Presence of AD-MSCs significantly enhanced bone regeneration for alveolar ridge augmentation. Scaffold-with-cell model exhibited better bone formation compared to scaffold-only models.
Yoshida et al. [229]	AD-MSCs from rats	Right inguinal region	Osteogenic-induced ADSC sheets	Rat	1 mm distal femur defect	Osteogenic-induced AD-MSC sheet may be more advantageous for bone healing than the AD-MSC sheet because of the higher number of osteocalcin-positive osteoblasts via the transplantation.
Wagner et al. [198]	AD-MSCs from human	Abdominoplastic surgery	Cancellous human bone allografts Human allogenic spongiosa chips	Rat	0.45 mm distal femur defect	Ratio of 84,600 cells per 100 mm^3^ scaffold is advantageous for vital cell population and cell seeding efficiency. Scaffolds seeded with AD-MSCs showed increased osteogenesis, proliferation, and angiogenesis, and elevated bone formation.
Liu et al. [184]	AD-MSCs from rats	Inguinal region	HDB	Rat	4 mm long bone defect	AD-MSCs-HDB has a strong osteogenic ability and successful regeneration of bone was found in segmental bone defects—a promising grafting material in bone tissue engineering.
Zanicotti et al. [207]	AD-MSCs from sheep	Hip region	Machined (MTi) and alumina-blasted (ABTi) titanium discs.	Sheep	10 mm × 9 mm × 7 mm femur epicondyle defect	AD-MSCs with titanium discs did not improve bone regeneration. Suspecting that the relative short duration given for healing (1 month) and the presence of titanium disc (1.5 mm thick) could have deteriorated bone healing process.
Chandran et al. [206]	AD-MSCs from sheep	Subcutaneous	SrHAcSrHA	Sheep	12 mm × 4 mm cortical bone defect	Local delivery of strontium and osteogenically induced AD-MSCs at the implant site facilitated improved osteogenesis and osteointegration.
González et al. [230]	AD-MSCs from canine	Subcutaneous	bTCP with/without fibronectin (Fn)	Canine	7 × 7 × 7 mm buccal cortical plate defect	bTCP coated with a combination of Fn and AD-MSCs appeared to encourage stabilization of the regenerated area, allowing a more efficient maintenance of the space at 3 months of healing.
Cowan et al. [40]	AD-MSCs and BM-MSCs	Subcutaneous anterior abdominal wall	PLGA	Rat	0.8 mm, 2 mm, 3 mm, 4 mm, and 5 mm parietal defect	AD-MSCs had higher proliferation capacity compared to BM-MSCs. Without any genetic manipulation or addition of exogenous growth factors, AD-MSCs were able to heal critical defects together with apatite-coated PLGA scaffolds at 12-week time point, filling 70–90% of area defect.
Carvalho et al. [202]	AD-MSCs from human	Lipoaspirates	Starch-polycaprolactone (SPCL) scaffolds	Rat	4 mm parietal bone defect	The use of AD-MSCs improved the outcomes of bone regeneration compared to the use of only scaffold alone. Wet-spun scaffolds were found to be biocompatible with AD-MSCs and facilitated bone regeneration.
Wu et al. [231]	AD-MSCs from human	Infrapatellar fat pad	Amniotic membrane (AM)	Rat	2.6 × 2.0 × 2.0 mm two-wall intrabony defect	AD-MSCs and AM co-culture system increased periodontal bone regeneration. Application of a co-culture system in periodontal disease is ideal due to its anti-inflammation, antiangiogenesis, and immunosuppression effects.
Zhang et al. [205]	AD-MSCs from rabbits	Bilateral epididymidesDorsal fat tissue	DCS DCS-CHA CHA	Rat	Transplanted into subcutaneous tissue	CHA enhanced the osteogenesis and blood vessel formation abilities of the DCS complexes in vivo. DCS complexes also promoted the osteogenesis and blood vessel formation potential of the CHA scaffold.

#### 3.2.2. AD-MSCs with Different Scaffold Materials

AD-MSCs have been seeded on various biomimetic scaffolds to evaluate their osteogenic ability and bone regenerative capacity when implanted into physiological conditions in vivo. Seeding AD-MSCs into these scaffolds has shown even more promising outcomes, reporting a higher osteogenic capacity than when scaffolds or AD-MSCs are implanted on their own. Generally, an ideal scaffold should facilitate optimal integration into the defect site and provide an osteoconductive environment, aiming to promote vascular inoculation and cellular ingrowth. Under this aspect, three properties of a scaffold, the material, structure, and porosity, must be carefully considered. The biomaterial has an ability to mimic the extra cellular matrix to facilitate hydroxyapatite formation and mineral deposition, and the structure to provide support for vascular formation and bone growth. The degree of porosity determines the interaction between the cell and the scaffold, where too small pores would result in a restricted area for the formation of vascular structures and cell migration, whereas too large pores would impair the surface area available for cell adhesion. A study found that the multilayer stacking of AD-MSCs seeded scaffolds with human endothelial progenitor cell (hEPC)-seeded tendon scaffolds, contributing to a significant enhancement of bone regeneration through enhanced vascularization formation. The nano-scale structures present on the scaffolds supersede those of the microscale, improving cellular alignment which resulted in enhanced vascularization accelerating regenerating effects, while also improving the differentiation and regenerative potential of AD-MSCs.

Several synthetic materials, derived from both inorganic and organic origins seeded with AD-MSCs, have been implanted onto defect sites in vivo to monitor the bone regeneration capacity. Yoon et al. [214] studied the osteogenic potential of AD-MSCs with PLGA scaffolds for bone regeneration in a rat critical-sized (8 mm) calvarial defect model. AD-MSCs were cultured in two different media, forming an undifferentiated group and a differentiated group, and discovered that constructs with a differentiated ADSC group showed more bone tissue regeneration than the undifferentiated AD-MSCs. The use of osteogenically differentiated AD-MSCs that have been cultured for at least 14 days exhibited a more robust bone regeneration capability when combined with PLGA implants. Semyari et al. [226] also seeded AD-MSCs into the PLGA, polyamide, and decellularized amniotic membrane for the treatment of calvarial defects (8 mm) in rabbits. Defect closure was observed in all cases 8 weeks post-operative, with the greatest defect closure in the animal treated with a polyamide scaffold. In the seeded groups, all markers (osteocalcin and osteopontin) showed significant enhancement obtained 4 and 8 weeks after implantation compared to the non-seeded groups.

Calcium phosphate ceramics (e.g., hydroxyapatite (HA), coralline-derived hydroxyapatite (cHA), tricalcium phosphate (TCP), calcium phosphate based cements, and bioglass) have great osteoinductive properties and have been utilized to treat bone defects [232,233]. Pourebrahim et al. [204] compared an autograft and stem cell graft (AD-MSCs + HA/bTCP) for the repair of maxillary alveolar cleft critical defects (5 mm) in a canine model. The integrity of the maxillary alveoli was restored in both test groups, but quantitatively more bone formation was found in the autografted site, whereas more collagen synthesis was observed in the stem cell grafted site that would eventually have a bone formation similar to the autografted side. This demonstrated that AD-MSCs-seeded scaffolds had good outcomes similar to an autograft—a golden standard for bone regeneration. Chandran et al. [206] discovered that strontium-incorporated HA using sheep AD-MSCs was able to enhance the osteogenic ability in mature lamellar bone formation for a critical-sized (12 mm × 4 mm) bone defect. The synergistic action of strontium and osteogenically induced AD-MSCs improved osteogenesis and osteointegration. Probst et al. [213] demonstrated that AD-MSCs with TCP-PLGA scaffolds significantly improved the osteogenic capacity with higher osteocalcin deposition at the defect sites. More new bone was formed in the large mandibular defect region in the seeded scaffold group as compared to the non-seeded group, quantified through micro-CTs calculating the total bone volume.

Alterative biomaterials using starch-polycaprolactone (SPCL) scaffolds seeded with undifferentiated AD-MSCs have been evaluated using rat models for the treatment of full-thickness cranial bone defects. The histological results showed that the presence of AD-MSCs improved the osteogenic function of SPCL scaffolds where the implant was better integrated and encapsulated by new tissues formed at the native defect interface. Although it is debatable if the scaffold was responsible for achieving these outcomes, it is definite that the presence of AD-MSCs enhanced bone regeneration [202]. A similar observation was reported by Lau et al. who loaded 3D-printed 80% PCL–20% tricalcium phosphate scaffolds with AD-MSCs in the form of SVF. The addition of cells to the scaffold significantly improved the bone volume in the alveolar ridge defect, led to a reduced immune response, and resulted in less fibrous encapsulation of the scaffold [108].

Maglione et al. [195] investigated the use of commercially available deproteinized bovine bone (Bio-Oss ^®^ Block, Geistlich, Wolhusen, Switzerland) and bovine cancellous granular with the addition of a collagen matrix to 10% (Bio-Oss Collagen ^®^, Geistlich, Wolhusen, Switzerland), with and without seeding AD-MSCs, for the treatment of a calvarial defect (6 mm). The micro-CT analysis revealed that all cases exhibited bone formation. The new bone formed with seeded AD-MSCs were found to be superposable with that obtained by the traditional regenerative technique, amplifying the cells’ potential. The findings indicate that the use of AD-MSCs combined with scaffolds accelerated osseous regeneration.

Although multiple studies have shown favorable outcomes of AD-MSCs in bone regeneration, others have expressed some concerns. Gonzalez et al. [230] reported that the use of AD-MSCs with bTCP for alveolar ridge defects (7 × 7 × 7 mm) did not have any advantage. However, with the introduction of fibronectin to the AD-MSCs + bTCP scaffold, the stabilization of the regeneration area was found to be beneficial for bone regeneration. The heterogeneity of experimental models in bone regeneration procedures using stem cells of different origins is an important drawback in the assessment of the advantages of the different scaffolds, as well as determining the most adequate construct for each type of defect. Di bella et al. [228] also conducted a study to monitor the effect of fibronectin in bone regeneration using a 15 mm rabbit calvarial defect model. They found that the surface treatment of fibronectin promotes bone formation within the scaffold due to its ability to modify the scaffold surface to be hydrophilic, thus enhancing adhesion, proliferation, and differentiation into the scaffold.

Bohnenblust et al. [234] also reported that the presence of osteogenic differentiated AD-MSCs did not increase the overall bone density with the non-seeded group, suggesting that the rationale is due to the calvarial bone being significantly larger than the concentration of seeded osteogenic differentiated AD-MSCs. Additionally, the duration given to monitor bone regeneration was only 6 weeks, which could be too short to observe bone growth. The duration given for bone healing is a critical determinant factor to access the bone regenerative capacities. Ovine AD-MSCs (oAD-MSCs) obtained from sheep used for femur defects (10 mm, 9 mm, and 7 mm, respectively) in a sheep model did not promote the regeneration of bone when implanted with titanium plates. The lack of increased matrix production was due to the medium characteristics and the varied ability of cells from different animals to differentiate into osteoblast-like cells. Additionally, only 1 month was given for the bone defect to regenerate, which could be insufficient time for regeneration to occur for a deep defect with the presence of a non-porous titanium plate that could have interfered with bone regeneration [207]. Despite the differences in the treatment of the cells or the scaffolds that are applied, one common feature reported by these different studies is that, overall, the usage of cells overcomes the results that the scaffold alone may provide.

#### 3.2.3. AD-MSCs as a Scaffold-Free Bone Regeneration Solution

Cell sheet technology, as a scaffold-free strategy, has been successfully applied to engineered bone fabrication [235]. Cell sheets are an alternative treatment method using tissue engineering to manage bone defects; mainly, it can be transplanted without a carrier and can serve as a scaffold at the defect site. Transplanted AD-MSCs sheets demonstrated bone healing in multiple in vivo models. Yoshida et al. [229] is the first to investigate the in vivo osteogenic ability of AD-MSCs sheets in an animal model. The capabilities of an AD-MSCs and osteogenic-induced AD-MSCs sheet to stimulate bone regeneration in a 1 mm defect on the distal femurs of 12-week-old rats were studied. ADSCs were cultured with a standard medium until they reached a stage of over confluence; subsequently, this was added into the medium and incubated for a week to achieve ADSC sheets. Osteogenic media were introduced to these sheets and cultured to obtain osteogenic-induced ADSC sheets. A higher degree of the osseous structure was formed and osteocalcin immunostaining showed that the content of the osteocalcin-positive osteoblast was higher in the osteogenic-induced ADSC sheet. Wang et al. [224] investigated the use of bidirectionally differentiated AD-MSCs for bone regeneration by using AD-MSCs to culture an osteogenic cell sheet and also to differentiate into endothelial progenitor cells (EPCs). The ADSC sheet–EPC complex was implanted subcutaneously into calvarial defects in rabbits. The ADSC sheet–EPC complexes formed dense and well-vascularized new bone tissue at 8 weeks after implantation, with a higher bone density in comparison to the control group. These sheets, due to their structure, did not require an additional scaffold nor could they be transported to defect sites by the minimally invasive method [236]. Osteogenic and angiogenic lineage-differentiated AD-MSCs enabled bone regeneration in vivo, but the lack of sufficient mechanical strength of the cell sheets and scaffolds made it difficult to control the shape of neo-mineralized tissues for the targeted formation of the bone. In view of this, using AD-MSCs sheets on a critical-sized defect might be a challenge.

Therefore, methods that combined AD-MSCs sheets with scaffolds were introduced. Zhang et al. [152] constructed double-cell sheets (DCS) by developing osteogenic cell sheets and vascular endothelial cell sheets using the induced culture of rabbit AD-MSCs for heterotic transplantation in rats. Coral hydroxyapatite (CHA) was combined with the DCS to form DCS–CHA complex. The presence of CHA expanded the effective space for cell and tissue growth. A favorable cell migration and connection was present, forming the stable blood vessel network. Mehrabani et al. [199] introduced AD-MSCs in the form of fibrin glue for the treatment of mandibular defects in a rabbit model. Comparing it with autografts, fibrin glue associated with AD-MSCs were able to regenerate new cortical bone with similar efficacy. Fibrin glue holds an upper hand in the treatment of mandibular defects as it can be easily located and shaped without occurring any inflammatory responses. Noting its osteogenesis potential with AD-MSCs, fibrin glue-AD-MSCs can be opted for as a new method for oral and maxillofacial surgery that is as good as the golden standard. Overall, for applications of AD-MSCs in vivo, it is important to consider if the cells can be administered safely into the surgical site and if the implanted cells are compatible with the scaffold to be functionally engrafted therein.

## 4. AD-MSCs in Clinical Applications

Multiple in vitro studies have shown promising results of AD-MSCs being able to differentiate into osteoblasts and chondroblast. Pre-clinical in vivo models have also demonstrated the successful regeneration and repair of bone defects through AD-MSCs. In clinical applications, autogenous AD-MSCs were harvested through subcutaneous/gluteal lipoaspirates and the buccal fat pad and seeded onto the scaffolds for implantation. The defects of calvarial, orthopedic, and oral–maxillofacial regions have been demonstrated to heal and improve the healing process with the use of AD-MSCs. Table 3 represents a summary of all the recent clinical studies conducted using AD-MSCs for bone regeneration.

Lendeckel et al. [237] reported that the first clinical trial was carried out with the use of AD-MSCs extracted from the gluteal area together with autologous cancellous bone grafts from the lilac crest. A clinical follow-up showed a symmetrical calvarial contour with no complications and found that the cancellous bone and macroporous sheets were in stable positions. Three-month-postoperative CT scans revealed marked ossification in the defect areas. Thesleff et al. [238] also reported the use of AD-MSCs in calvarial defects in four patients who underwent cranioplasty (Figure 5). Subcutaneous abdominal fats were harvested and seeded with beta-tricalcium phosphate (betaTCP) granules. The capacities of AD-MSCs and the osteoconductiveness of betaTCP acted synergistically, producing a well-ossified construct. There were no post-operative complications and good outcomes in ossification were exhibited. A six-year clinical follow-up showed no adverse events and all patients recovered from the surgeries [212].

**Figure 5 ijms-25-06805-f005:**
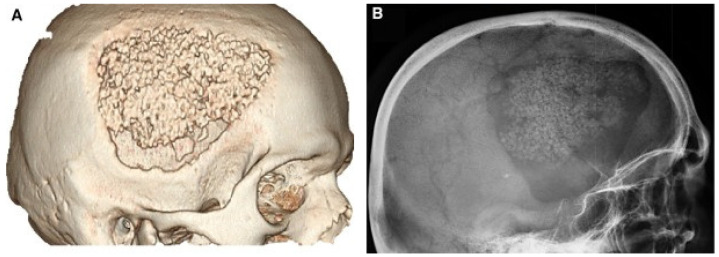
Good clinical outcomes for calvarial defect treatment with AD-MSCs and betaTCP granules—forming ossification. (**A**) Reformatted CT image of the head 13 months post-operatively. Partial resorption of the graft is seen at the basal area. (**B**) Skull X-ray 6 years post-operatively shows substantial resorption of the graft at the borders. Reproduced with permission from Thesleff et al., Cranioplasty with Adipose-Derived Stem Cells, Beta-Tricalcium Phosphate Granules and Supporting Mesh: Six-Year Clinical Follow-Up Results. Stem Cells Transl Med 2017, 6, 1576–1582, Figure 2 (https://doi.org/10.1002/sctm.16-0410) under the terms and conditions of the Creative Commons Attribution (CC BY) license (http://creativecommons.org/licenses/by/4.0/ accessed on 4 October 2023) [238].

AD-MSCs have also been used for the treatment of orthopedic conditions. Pak et al. [239] described the use of AD-MSCs in the form of the SVF for the treatment of 91 patients with various orthopedic conditions. Based on the patient’s visual analogue scale (VAS), the treatment improved by 50–60% with no neoplastic complications, showing that the method provides a safe, long-term pain-improving alternative.

Saxer et al. [215] used the combination of the SVF with ceramic granules for the treatment of low-proximal humeral patients in eight patients. Biopsies of the repair tissue after 12 months showed the formation of bone ossicles, distinct from osteo-conducted bone, and exhibited good recovery. The freshly isolated human SVF cells could form blood vessels and de novo bone tissues in the absence of any in vitro priming or of exogenously supplemented morphogenetic stimuli. Dufrane et al. [240] designed a 3D AD-MSCs graft. Without the presence of a scaffold, bone restoration was observed without any side effects, promoting osteogenesis even in extreme conditions of non-union bone with minor donor site morbidity and no oncological side effects. Veriter et al. [241] also tested the use of a scaffold-free osteogenic 3D ADSC graft for the treatment of non-union bone (congenital pseudarthrosis/intercalary bone allograft implantation after tumor resection) and presented osteocalcin expression, the synthesis of the extracellular matrix, and mineralization in each condition.

Bone regeneration in the oral and maxillofacial region is a challenge due to multiple physiological structures such as sensory organs, facial skeletal features, cartilage, and blood vessels. Additionally, the oral region is more prone to bacterial contamination, requiring a more precautious approach for regenerative efforts [242]. AD-MSCs isolated from the buccal fat pad are a commonly adopted harvest source for bone regeneration in the oral/maxillofacial region and have shown successful regeneration outcomes. Khojasteh [196] explored the extraction of AD-MSCs from the buccal fat pad in the treatment of alveolar cleft defects in 10 patients. When used in combination with natural bovine bone mineral granules, successful healing with no fistula or oronasal communication was observed in all cases except one that developed partial dehiscence, which was managed by prescribing mouthwash. A histological analysis showed new lamellar bone with an osteoblastic rim without inflammatory cell infiltration. Mesimaki et al. [208] reported a successful maxillary reconstruction with a microvascular flap using AD-MSCs seeded in beta-tricalcium phosphate (bTCP) with bone morphogenetic protein-2 (BMP-2). The patient had a large bony defect due to a large recurrent keratocyst and had to undergo a hemi maxillectomy. After 4 months postoperative, the implants osseo-integrated without any side-effects and the regeneration of the palatal mucosa was complete. Sandor et al. [209] also showed that AD-MSCs with the combination of bTCP and BMP-2 successfully reconstructed large mandibular defects (10 cm) without the need for ectopic bone formation.

**Table 3 ijms-25-06805-t003:** Selected clinical studies of AD-MSCs for bone tissue engineering, grouped according to experimental model.

First Author	Cell Source and Type	Scaffold/Fillers Used	Harvest Method	Experimental Model	Model Details	Site of Injury/Area of Reconstruction	Key Findings
Lendeckel et al. [237]	AD-MSCs	Macroporous sheets	Left gluteal area	Calvarial Defect	Due to the limited amount of autologous cancellous bone, AD-MSCs were applied	7-year-old girl with multifragment calvarial fractures	Clinical follow-up has shown symmetrical calvarial contour. There were no neurological deficits nor pathological findings. CT scans 3 months postoperatively showed a marked ossification in the defect areas.
Thesleff et al. [212]	AD-MSCs	bTCP granules	Approx. 100 mL subcutaneous abdominal fat	Calvarial Defect	5 patients who underwent cranioplasty	The average defect size was 8.1 × 6.7 cm^2^	No clear outcomes were reported to show that AD-MSCs with bTCP granules improved ossification or bone regeneration, possibly due to infection and tumor recurrence found. Regardless, no adverse events were reported and all patients recovered.
Thesleff et al. [238]	AD-MSCs	bTCP granules	200 mL of subcutaneous abdominal fat	Calvarial Defect	4 patients who underwent cranioplasty	HemangiomaFrontal meningiomaAcute subdural hematomaCranial base meningioma	Capacities of AD-MSCs and the osteoconductivity of the bTCP act synergistically towards producing a well-ossified construct, regenerating bone in adult critical-size calvarial defects
Pak et al. [239]	SVF, AD-MSCs with platelet-rich plasma (PRP)	No Scaffold	100 mL of lipoaspirates from lower abdomen	Orthopedic Defect	Mixture of AD-MSCs and PRP were percutaneously injected into knees, hips, low backs, and ankles.	15 avascular necrosis 7 hip osteoarthritis 74 knees 2 low-back spinal disc herniation	100 joint injections of AD-MSCs, in the form of SVF, with PRP shows that AD-MSCs/PRP treatment is safe and provides long-term pain improvement. No evidence of neoplastic complications in any implant sites in 91 patients with 100 joints.
Dufrane et al. [240]	AD-MSCs	No Scaffold	1.926 g of fatty tissue by subcutaneous biopsy	Orthopedic Defect	AD-MSCs in osteogenic media, supplemented with demineralized bone matrix.	3 patients with bone non-unions due to congenital pseudarthrosis or acquired pseudarthrosis	The final bone formation was stable and did not rupture with forceps manipulation nor had donor site morbidity. No acute side-effects associated with the graft up to 4 years after transplantation.
Vériter et al. [241]	AD-MSCs	No Scaffold	9.7 ± 13.7 g by lipoaspiration	Orthopedic Defect	Assess the safety and efficacy of AD-MSCs (1) in bone non-union and (2) in dermal reconstruction of non-healing chronic wounds.	17 patients who had not experienced any success with conventional therapies	3D osteogenic-like structure allowed bone consolidation for up to 4 years without any notable complications in oncologic patients with tumor resection. No serious adverse events were found (up to 54 months).
Saxer et al. [215]	SVF, AD-MSCs	Ceramic granules within fibrin gel	Lipoaspirate	Orthopedic Defect	Evaluate the efficacy of SVF progenitors at bone fracture site.	Patients with low-energy proximal humeral fractures in 8 patients (64–84 years old) along with standard open reduction and internal fixation	Biopsies of the repair tissue (after up to 12 months), upon plate revision or removal, exhibited formation of bone ossicles, structurally disconnected and morphologically distinct from osteoconducted bone, suggesting the osteogenic nature of implanted SVF cells.
Mesimaki et al. [208]	AD-MSCs	Titanium bTCP	200 mL of subcutaneous abdominal fat	Oral– Maxillofacial Defect	Evaluate the method to reconstruct a major maxillary defect using AD-MSCs with recombinant human protein (rhBMP) and a scaffold.	65-year-old male, who had undergone a hemi-maxillectomy 28 months earlier due to a large recurrent keratocyst	After 8 months of follow-up, the flap had developed mature bone structures and vasculature and was transplanted into the defect area. The combination of AD-MSCs with bTCP acted synergistically producing a well-ossified construct.
Sandor et al. [209]	AD-MSCs	bTCP granulesTitanium mesh	200 mL of subcutaneous adipose tissue	Oral– Maxillofacial Defect	To access the reconstruction of large anterior mandibular defect using 1-stage in situ bone formation instead of multistep ectopic bone formation.	Replacement of a 10 cm anterior mandibular ameloblastoma resection defect, reproducing the original anatomy of the chin	AD-MSCs in combination with bTCP and BMP-2 successfully treated large mandibular defect without the need for ectopic bone formation; in situ ossification was present which saved the patient a second surgical site as no vessel anastomosis or second transplantation step was necessary.
Sandor et al. [210]	AD-MSCs	Bioactive glassbTCP	50–200 mL of subcutaneous adipose tissue	Oral– Maxillofacial Defect	Access the efficacy of AD-MSCs seeded onto resorbable scaffold materials for subsequent reimplantation into hard-tissue defects.	13 consecutive cases of cranio-maxillofacial hard-tissue defects at four anatomically different sites—frontal sinus (3 cases), cranial bone (5 cases), mandible (3 cases), and nasal septum (2 cases)	Healed hard-tissue grafts in the defect area were functioning according to the demands of their new native sites during the healing period. Resorption of the constructs was more than expected in the cranial defect. For cranial defect, a sturdy, non-resorbable material should be used to cater to the dural pulsation that is exposed to the cranial wounds to provide optimal healing environment.
Prins et al. [243]	SVF, AD-MSCs	Straumann dental implants	150 mL by lipoaspriate	Oral– Maxillofacial Defect	SVF applied in a one-step surgical procedure with calcium phosphate ceramics (CaP) to increase maxillary bone height for dental implantations.	10 patients were included in this study, who were partially edentulous in the posterior maxilla	The bioactive implants generated showed successful one-step surgical implantation and healing outcomes. The osteoid percentages were higher at the AD-MSCs’ seeded group than unseeded group. SVF improved bone formation capacity, resulting in higher bone volume.
Khojasteh et al. [196]	AD-MSCs	Autograft	3 to 5 mL from buccal fat pad	Oral– Maxillofacial Defect	Combination of different grafts with AD-MSCs derived from BFC was accessed for the restoration of unilateral cleft lip and palate.	Ten patients with unilateral cleft lip and palate	Complex of AD-MSCs and scaffold improved bone regeneration with lower donor site morbidity and recovery duration. Combining AD-MSCs with AIC-enhanced new bone growth and LRCP was found to be useful in housing scaffolds loaded with AD-MSCs.
Cardiel et al. [211]	AD-MSCs	Compression titanium plate	50 cc adipose tissue from abdominal region	Oral– Maxillofacial Defect	Treatment of defect with/without AD-MSCs	16 patients with mandible condyle fractures or associated facial fractures	Similar ossification values were obtained after 4 weeks when the use of AD-MSCs was compared to simple fracture reduction. However, after 12 weeks, the AD-MSCs group had a 36.48% higher ossification rate.
Guasch et al. [191]	SVF, AD-MSCs	bTCP biphasic calcium phosphate carriers	>125 mL from lipoaspiration	Oral– Maxillofacial Defect	SVF seeded on two types of calcium phosphate carriers, were accessed to understand their potential for bone regeneration.	10 patients with and used for Maxillary Sinus Floor Elevation Model in a one-step surgical procedure	Increase in the quantity and maturity of blood vessels was found, particularly near the cranial regions. Bone percentages are proportional to blood vessel formation and are higher in AD-MSCs seeded group in bTCP-treated patients. SVF was found to have pro-angiogenic bone formation-enhancing effects.
Mazzoni et al. [203]	AD-MSCs	HA–collagen hybrid	Purchased	Oral–Maxillofacial Defect	In vitro and in vivo analysis of HA–collagen scaffold effectiveness for bone regeneration.	50 patients with zygomatic augmentation and bimaxillary osteotomy	Presence of mature bone was found pre-eminently at the periosteal side, whereas the presence of new immature bone was detected entirely in the deep layer of the native bone. Successful clinical outcome was found in patients, showing significant osteogenic induction.

Thirteen consecutive cases of cranio-maxillofacial hard tissue defects at four defects of different sizes, namely frontal sinus, cranial bone, mandible, and nasal septum reconstruction, was treated using AD-MSC-seeded scaffolds (either bioactive glass or bTCP) with or without the addition of BMP-2. The successful integration of the implanted construction was observed in all cases except for the cranial defects where the resorption of the constructs was more than expected. Sustaining the constant dural pulsation is necessary for the management of cranial defects, thus the use of a rigid structural material could be an alternative that can be considered [210]. In the 13 cases, the ectopic muscle pouch site was used for bone induction, differing from other mandibular and maxillary reconstruction studies [209]. With this, there was no need for an ectopic bone formation step, reporting the sufficient availability of soft tissue to cover the combination scaffold, simplifying the overall protocol. Such a technique has been called in situ bone formation and it saves the patient from a second surgical site and a major surgical procedure because no vessel anastomosis or second transplantation step is necessary [210].

Prins et al. [243] reported the first in-human study using AD-MSCs from SVF seeded on either bTCP or biphasic calcium phosphate carriers for maxillary sinus floor elevation for dental implant placement. Six months postoperatively, the biopsies indicated that the bone and osteoid percentages were higher in all groups compared to the control, suggesting that the presence of AD-MSCs seeding has significantly enhanced the bone-forming capacity in the augmented area and may result in a higher bone volume following dental implant placement. Cardiel et al. [211] dealt with mandible condyle fractures by applying AD-MSCs with a compression titanium plate. CT images at week 12 showed a 36.48% higher ossification rate.

Through the clinical models, we observe that bone tissue engineering techniques that support adequate vascularization are important to provide the timely and adequate transport of nutrients, facilitate proper waste removal, and to supply progenitor cells for tissue remodeling and repair. Multiple studies have shown that good vascularization supports better bone regenerations and that vascularization precedes osteogenesis during bone growth [160]. Poor angiogenesis is a common and critical obstacle for bone tissue regeneration, where the regeneration of tissue over 200 μm supersedes the capacity of the nutrient supply and waste removal from the tissues, which would require an extensive supply of vascular networks [40]. Therefore, the use of angiogenic growth factors and/or the transplantation of proangiogenic cells, such as endothelial progenitor cells (EPCs) with scaffolds, are commonly adopted. Using angiogenic growth factors and/or transplanting the proangiogenic cells also encompassed disadvantages, since perivascular cells such as mural cells are inhibitory for the formation of native multi-layered mature blood vessels [244,245]. Therefore, the potential of AD-MSCs to stimulate angiogenesis holds interesting promises for the field of tissue engineering where the regeneration of bone seemed to correlate with blood vessel formation. Guasch et al. [191] evaluated the vascularization in relation to the bone formation potential of AD-MSCs seeded on calcium phosphate carriers for maxillary sinus floor bone augmentation. Higher blood vessel counts and the presence of more mature vessels (with a mean diameter of 30 um) were found in the area of bone formation, which were well above the size of capillaries (5–10 um). These areas were treated with AD-MSCs and the bone formations were more significant. Alternatively, the upregulation of osteogenic genes were found to improve matrix mineralization and cell viability, enhancing bone growth. Preparing the recipient site by maximizing the contact of the seeded scaffold with a region of muscles that is well vascularized is important for bone regeneration. Mazzoni et al. [203] showed that the continuous expression of osteogenic, osteoclastic, and chondrogenic genes favored bone regrowth in patients whom underwent malar augmentation procedures using hybrid scaffold collagen/Pro Osteon. Follow-up after 3 years showed an almost complete radiopacity and apparent corticalization of the bone in contact with the scaffold construct, demonstrating significant osteogenic induction. Particularly, the expression of protein CD56 was directly proportional to new bone formation and the protein was in agreement with the osteogenic genes that were found to be upregulated in the cellular model of stem cells. HA–collagen with AD-MSCs demonstrates promising outcomes to be used in remodeling and bone regrowth.

Another way to tackle the problem of poor angiogenesis is to review the mechanisms of bone development, namely endochondral ossification (ECO) and intramembranous ossification (IMO) (mentioned in Section 2.4). In the past three decades, the field of bone tissue engineering is predominantly based on the direct osteogenic differentiation of MSCs with the goal of forming a bone-like matrix, mimicking the IMO pathway [88]. Compared to ECO, the simpler and better-understood mechanism of the IMO pathway makes the IMO process easier and faster to carry out experimentally [246]. Despite achieving major progress, IMO strategies have some limitations, the major limitation being the lack of a functional vascular supply and the formation of a necrotic avascular core due to the inhibition of the vascular ingrowth by the newly formed mineralized matrix [247]. This limitation makes IMO ineffective for large bone defects. To address such limitations, researchers are gradually adopting an alternative strategy mimicking the ECO process, where MSCs are first differentiated into chondrocytes to form a cartilaginous matrix. After promoting angiogenesis and vascular ingrowth in the cartilaginous matrix, the MSCs are induced to undergo osteogenic differentiation to replace the cartilaginous matrix with mineralized bone tissues [88,248]. This approach, also known as endochondral bone engineering, overcomes problems associated with poor vascularization because (a) chondrocytes can resist hypoxia and survive much better in an avascular environment than MSCs and osteoblasts [249], (b) hypertrophic chondrocytes can induce angiogenesis and ossification by releasing factors such as VEGF and BMPs [250,251], and (c) endochondrally primed grafts have been shown to integrate faster with host tissues after implantation in vivo [247,252]. Currently, endochondral bone engineering is still in the early stages of development, with most studies still using bone marrow stem cells and in vivo studies still using small animal models such as mouse, rat, and rabbit [88,247,253]. The field of endochondral bone engineering, especially the use of AD-MSCs and the translation of the ECO approach into larger animal models and humans offers significant potential for further exploration.

The numerous developments in the field of bone tissue engineering have significantly contributed to the validation of novel strategies as viable treatment options for the reconstruction of challenging bone defects. In the case of oral and maxillofacial defects, mechanical and structural supports are key factors that are required for reconstruction due to frequent movements of the jaw from speech and masticatory functions [242]. The reported clinical studies confirmed the findings obtained from the in vitro and in vivo models, suggesting that native or cultured AD-MSCs, when used alone or in combination with biomimetic scaffolds with or without growth factors, are effective in stimulating bone healing.

## 5. Conclusions

In the event of critical-sized defects, diseases, or old age, bone regeneration is significantly impaired and would require external intervention for healing to occur. The therapeutic strategies aimed at regenerating bone have evolved, introducing new techniques for reconstructive surgery, which have significantly improved clinical outcomes.

AD-MSCs have demonstrated their potential as a new promising tool for regenerative applications because of their low immunogenicity and their ability to differentiate into multiple lineages and secrete various cytokines. Over the past few decades, the emerging adoption of AD-MSCs in tissue engineering techniques has answered multiple clinical problems in bone regeneration. The advantage of obtaining a large quantity of AD-MSCs via a minimally invasive harvest method makes them a promising stem cell source. Methods using bioactive proteins alone or loading bioactive proteins on scaffolds, and alternative materials with or without cells, are moderately effective. The ability of undergoing osteogenic differentiation without any stimulation when seeded on an osteoconductive scaffold in vivo makes AD-MSCs a promising candidate for bone tissue engineering. Many studies have demonstrated that AD-MSCs combined with a scaffold have a significantly enhanced and accelerated bone formation. Extracellular biomimetic scaffold materials not only provided structural support, but supplicated cells with an area of adhesion, proliferation, matrix maturation, mineralization, and angiogenesis for tissue formation.

The vasculature is a key component responsible for the transport of nutrients and oxygen for bone formation. Creating an optimal environment to promote vascularization is thus critical in tissue engineering. The material, pore size, and structural morphology of the scaffold together with providing a hypoxic environment should also be carefully considered. Switching from an intramembranous ossification approach to an endochondral ossification approach can also overcome the issues associated with vascularization. Overall, bone tissue engineering using AD-MSCs has significantly contributed to treating multiple critical-sized defects, highlighting the promising future of AD-MSCs for bone regeneration.

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
