# Peer review of "Role of Adipose-Derived Mesenchymal Stem Cells in Bone Regeneration"

_ijms, 2024, doi:10.3390/ijms25126805_

Round 1
Reviewer 1 Report
Comments and Suggestions for Authors
In this review, the authors comprehensively summarized the therapeutic potential of AD-MSCs for treating bone defects, primarily focusing on in vitro, in vivo, and clinical studies. I appreciate the detailed work involving discussions and summarization from lab to bench. However, there are significant aspects that have not been covered, which could be crucial in this field. Therefore, I recommend a major revision with the following suggestions:
1. What are the primary mechanisms through which AD-MSCs facilitate bone repair? Is it predominantly through endochondral ossification or intramembranous ossification? This distinction is critical for understanding the therapeutic potential and application of AD-MSCs in bone repair.
2. The authors mentioned extracellular vesicles (EVs) as an important secretion in the abstract. This topic is not adequately expanded upon in the review. It is highly recommended that the authors include a detailed discussion on EVs, specifically highlighting which bioactive components in EVs from AD-MSCs contribute to bone repair. (e.g., mRNAs: doi.org/10.1002/advs.202302622; miRNAs: doi.org/10.3389/fcell.2021.833840)
3. The applications of AD-MSCs in cell-based therapy and the applications of their secretions (e.g., EVs, growth factors) in cell-free therapy should be compared. Determining which approach is more effective is crucial for advancing the field and should be a focal point in this review.
4. Session 3. AD-MSCs in Pre-Clinical Applications. The section on pre-clinical applications of AD-MSCs appears to be somewhat disorganized and lacks logical flow. It is recommended that the authors restructure this section to categorize the application of AD-MSCs therapy by different regions, such as oral, tibial fracture, spinal cord, etc., rather than grouping them into in vitro and in vivo studies. This would enhance the clarity and utility of the review.
Comments on the Quality of English LanguageMinor editing of English language required
Author Response
First of all, we would like to thank the reviewer for taking the time to review this manuscript and offer valuable feedback and suggestions. Please find the detailed responses below and the corresponding revisions highlighted in yellow in the re-submitted file. (The revisions highlighted in green are made according to the other reviewer's feedback.)
Comment 1:
What are the primary mechanisms through which AD-MSCs facilitate bone repair? Is it predominantly through endochondral ossification or intramembranous ossification? This distinction is critical for understanding the therapeutic potential and application of AD-MSCs in bone repair.
Response 1:
We thank the reviewer for the comment. We have modified section 2.4 (section 2.3 in the previous version) to include explanations on the mechanism of bone development and bone fracture healing, as well as endochondrous ossification and intramembranous ossification. We have also added one paragraph in Section 4 (before the last paragraph) and one sentence in Section 5 to discuss the limitations of intramembranous ossification strategies and the potentials of endochondrous ossification strategies to overcome the problem of poor vascularization in AD-MSC-seeded defects.
Comment 2:
The authors mentioned extracellular vesicles (EVs) as an important secretion in the abstract. This topic is not adequately expanded upon in the review. It is highly recommended that the authors include a detailed discussion on EVs, specifically highlighting which bioactive components in EVs from AD-MSCs contribute to bone repair. (e.g., mRNAs: doi.org/10.1002/advs.202302622; miRNAs: doi.org/10.3389/fcell.2021.833840)
Response 2:
We thank the reviewer for the comment. We have added a new section (Section 2.7.3) to cover the discussions on EVs.
Comment 3:
The applications of AD-MSCs in cell-based therapy and the applications of their secretions (e.g., EVs, growth factors) in cell-free therapy should be compared. Determining which approach is more effective is crucial for advancing the field and should be a focal point in this review.
Response 3:
We thank the reviewer for the comment. We have added a new section (Section 2.7) to cover the AD-MSC secretions and discussions on the various types of secretions.
Comment 4:
Session 3. AD-MSCs in Pre-Clinical Applications. The section on pre-clinical applications of AD-MSCs appears to be somewhat disorganized and lacks logical flow. It is recommended that the authors restructure this section to categorize the application of AD-MSCs therapy by different regions, such as oral, tibial fracture, spinal cord, etc., rather than grouping them into in vitro and in vivo studies. This would enhance the clarity and utility of the review.
Response 4:
We thank the reviewer for the comment. After internal discussion among our team, we have decided to keep the grouping for in vitro and in vivo studies. Nevertheless, we have improved Table 2 and Table 3 by sorting the studies according to the experimental model (e.g. calvarial defect, orthopedic defect, etc). We have also improved the organisation and flow of Section 3 by introducing new headers that could help in making the content more coherent.
Namely :
3.1 In-vitro applications
3.1.1 AD-MSCs as a cell source for bone regeneration
3.1.2 AD-MSCs from the oral region for oral-maxillofacial applications
3.1.3 AD-MSCs and the role of scaffolds
3.2. In-vivo applications 3.2.1 AD-MSCs response in bone regeneration.
3.2.2 AD-MSCs with different scaffold materials
3.2.3 AD-MSCs as a scaffold-free bone regeneration solution We would like to thank the reviewer for taking the time to review our revisions. We appreciate your valuable feedback to our revisions.

Reviewer 2 Report
Comments and Suggestions for Authors
Comments to the Authors
The manuscript review ijms-2989093 titled “Role of Adipose-Derived Mesenchymal Stem Cells in Bone Regeneration” focuses on an interesting field, describing a detailed overview of the state art of mesenchymal stem cells derived from adipose tissue and their features in bone regeneration.
The review includes many relevant and recent references. It covers various aspects related to the biological and functional characteristics of the AD-MSC cells, providing well-presented examples in vitro and in vivo.
However, some revisions are recommended.
Title: Role of Adipose-Derived Mesenchymal Stem Cells in Bone Regeneration
Lines 99-101: The primary objective of this work is to showcase AD-MSCs as an example of adult stem cells that can be easily derived from the abundant source of adipose tissue and used for their therapeutic efficacy. These cells exemplify “adult mesenchymal stem cells, crucial in cell therapy and regenerative medicine due to their self-renewal ability, pluripotent differentiation ability, easy accessibility, and exceptional genomic stability” It is suggested to accurately attribute multipotent ability to adult stem cells, as it is a more appropriate term than pluripotent.
Lines 102-103: Furthermore, considering the different sources of adult mesenchymal stem cells, I suggest including mesenchymal stem cells derived from follicular fluid of ovarian follicles in the list.
The Authors discuss the induction of stem cells suitable for differentiation and could reference studies that distinguish AD-MSC from dedifferentiated cells derived from mature adipocytes.
The review includes five figures, modified from five papers in this field, as the legend reports. However, I believe it will be necessary to mention that permission to modify and publish has been requested from the original journals. I suggest that the Authors add this information.
English can be improved by removing the numerous repetitions that appear everywhere.
Minor revision concerns:
- Figure 1b: BM-MSC
- Line 130: insert spacing between“engineering by”
- Line 352-361: references?
- Line 479: Please, report into the legend of Figure “Unrestricted somatic stem cells (USSCs)”.
Anyway, I can conclude that this manuscript is suitable for publication in the International Journal of Molecular Sciences after the recommended revisions.
Comments on the Quality of English LanguageEnglish can be improved by removing the numerous spelling repetitions that appear everywhere.
Author Response
First of all, we would like to thank the reviewer for taking the time to review this manuscript and offer valuable feedback and suggestions. Please find the detailed responses below and the corresponding revisions highlighted in green in the re-submitted file. (The revisions highlighted in yellow are made according to the other reviewer's feedback.)
Comment 1:
Lines 99-101: The primary objective of this work is to showcase AD-MSCs as an example of adult stem cells that can be easily derived from the abundant source of adipose tissue and used for their therapeutic efficacy. These cells exemplify “adult mesenchymal stem cells, crucial in cell therapy and regenerative medicine due to their self-renewal ability, pluripotent differentiation ability, easy accessibility, and exceptional genomic stability” It is suggested to accurately attribute multipotent ability to adult stem cells, as it is a more appropriate term than pluripotent.
Response 1:
We thank the reviewer for the comment. We have changed the term to multipotent.
Comment 2:
Lines 102-103: Furthermore, considering the different sources of adult mesenchymal stem cells, I suggest including mesenchymal stem cells derived from follicular fluid of ovarian follicles in the list.
Response 2:
We thank the reviewer for the comment. We have included ovarian follicular fluid and a few other types of tissues. We also replaced references 29 & 30 with newer articles to reflect more recent trends.
Comment 3:
The Authors discuss the induction of stem cells suitable for differentiation and could reference studies that distinguish AD-MSC from dedifferentiated cells derived from mature adipocytes.
Response 3:
We thank the reviewer for the comment.
Comment 4:
The review includes five figures, modified from five papers in this field, as the legend reports. However, I believe it will be necessary to mention that permission to modify and publish has been requested from the original journals. I suggest that the Authors add this information.
Response 4:
We thank the reviewer for the comment. We have modified the figure captions to state that the figures are “reproduced with permission from” the original article “under the terms and conditions of the Creative Commons Attribution (CC BY) license”.
Comment 5:
English can be improved by removing the numerous repetitions that appear everywhere.
Response 5:
We thank the reviewer for the comment. We have made our best effort to improve the English but if we are not sure whether we have addressed your concerns about the “numerous repetitions”. If you can provide a few examples of the “numerous repetitions”, we can go through the manuscript again to make the necessary improvements.
Minor revision concerns:
- Figure 1b: BM-MSC
- Done
- Line 130: insert spacing between“engineering by”
- Done (Line 131 in new version)
- Line 352-361: references?
- Done (Line 576-581 in new version)
- Line 479: Please, report into the legend of Figure “Unrestricted somatic stem cells (USSCs)”.
- Done (Line 542 in new version)
We would like to thank the reviewer for taking the time to review our revisions. We appreciate your valuable feedback to our revisions.

Round 2
Reviewer 1 Report
Comments and Suggestions for Authors
The authors have addressed my concerns.